

# Characterizing Southeast Greenland fjord surface ice and freshwater flux to support biological applications

Twila A. Moon[1*], Benjamin Cohen[2*], Taryn E. Black[2,3], Kristin L. Laidre[2], Harry Stern[2], Ian Joughin[2]

[1]National Snow and Ice Data Center, Cooperative Institute for Research in Environmental Sciences, University of Colorado, Boulder, 80309, USA
[2]Polar Science Center, Applied Physics Laboratory, University of Washington, Seattle, 98105, USA
[3]Earth System Science Interdisciplinary Center, University of Maryland, College Park, MD 20742, USA
* These authors contributed equally to the work.

*Correspondence to*: Twila A. Moon (twila.moon@colorado.edu)

**Short summary.** *(500-character limit)*

The complex geomorphology of Southeast Greenland (SEG) creates dynamic fjord habitats for marine top predators, with glacier-derived floating ice, pack and landfast sea ice, and freshwater flux. We investigate the SEG fjord physical environment, with focus on surface ice conditions, to provide a regional characterization to support biological research. As Arctic warming continues, SEG may serve as a long-term refugia for ice-dependent wildlife due to projected regional ice sheet persistence.

**Abstract.**

Southeast Greenland (SEG) is characterized by complex morphology and environmental processes that create dynamic habitats for resident marine top predators. Active glaciers producing solid ice discharge, freshwater flux, offshore sea ice transport, and seasonal landfast ice formation all contribute to a variable, transient environment within SEG fjord systems. Here, we investigate a selection of physical processes in SEG to provide a regional characterization to reveal physical system processes and support biological research. SEG fjords exhibit high fjord-to-fjord variability regarding bathymetry, size, shape, and glacial setting, influencing some processes more than others. For example, the timing of offshore sea ice formation in fall near SEG fjords progresses temporally southward across latitudes while the timing of offshore sea ice disappearance is less dependent on latitude. Rates of annual freshwater flux into fjords, in contrast, are highly variable across SEG, with annual average input values ranging from ~$1\times10^8$ m³ to ~$1.25\times10^{10}$ m³ (~0.1–12.5 Gt) for individual fjords. Similarly, rates of solid ice discharge in SEG fjords vary widely – in part due to the irregular distribution of active glaciers across the study area (60°N-70°N). Landfast sea ice, assessed for 8 focus fjords, is seasonal and has a spatial distribution highly dependent on individual fjord topography. Conversely, glacial ice is deposited into fjord systems year-round, with the



spatial distribution of glacier-derived ice dependent on glacier termini location. As climate change continues to affect SEG,
the evolution of these metrics will be individually variable in their response, and next steps should include moving from
characterization to system projection. Due to projected regional ice sheet persistence that will continue to feed glacial ice
into fjords, it is possible that SEG could remain a long-term (century to millennia scale) refugia location for polar bears and
other ice-dependent species, demonstrating a need for continued research on the SEG physical environment.

## 1 Introduction and motivation

Rapid changes across the Greenland coastal environment are influencing the linked physical and biological fjord systems.
The Greenland Ice Sheet and peripheral glaciers and ice caps are undergoing substantial retreat along marine- and land-
terminating boundaries, revealing new ocean and terrestrial zones (Moon et al., 2020; Kochtitzky and Copland, 2022;
Bosson et al., 2023). For some marine-terminating glaciers, changing ice dynamic and terminus locations are altering iceberg
calving styles or rates (e.g., van Dongen et al., 2021), with potential influence on glacier-derived fjord ice that forms
important habitat for polar bears *(Ursus maritimus),* seals, and many other marine species (e.g., Laidre et al., 2022).
Increases in ice sheet surface melt are also changing the timing and quantity of subglacial meltwater discharge and terrestrial
riverine freshwater input into the coastal fjords (e.g., van As et al., 2018). Depending on the fjord bathymetry and glacier
grounding line depth, this subglacial discharge may entrain deeper nutrient-rich ocean water and assist in redistributing it to
the surface photic zone to support enhanced productivity (Hopwood et al., 2018; Meire et al., 2023) or alter the ecosystem in
other potentially significant ways (e.g., Hawkings et al., 2021; Hopwood et al., 2020). Additional terrestrial runoff adds to
coastal zone freshwater (e.g., from Norway: McGovern et al., 2020), though impacts are less well documented for Greenland
(Meire et al., 2023). Despite the rapid changes underway, progress is still needed on fundamental physical characterization
of the Greenland coastal zone, including the remote Southeast Greenland (SEG) region (Fig. 1).
Earlier work characterized the landfast sea ice and glacier-derived fjord surface ice for five SEG fjords that were biologically
relevant to polar bears (Laidre et al., 2022). This research revealed that glacier-derived fjord surface ice exists during time
periods outside of the landfast sea ice season, and that this glacier-derived ice can act as an alternative habitat platform for
marine species, allowing small populations to persist in areas they may not otherwise be able to. Motivated by the biological
insight enabled via enhanced physical system knowledge, here we extend our characterization of the SEG fjord physical
environment. Examining the full SEG region of interest (Fig. 1), we describe the freshwater flux, offshore sea ice, and solid
glacier ice discharge behavior across the region during 2015 through 2019. We also expand from the five focus fjords used
in Laidre et al. (2022) to eight focus fjords across SEG (Fig. 1, Table 1). For these focus fjords, we analyze landfast sea ice
and glacier-derived ice presence in time and space and compare these results with offshore sea ice from satellite





Figure 1. Southeast Greenland region of study, showing the 52 fjord systems defined across the full region (blue shading) and the 8 focus fjords used for fast ice and glacier-derived ice analysis (pink outlines). Locations of outlet glaciers considered in analysis of solid ice discharge are shown (green points).

observations, and freshwater flux, sea surface temperature, and sea ice cover from a regional climate model. Our results are
designed to expand knowledge of SEG fjord environments and pair with ongoing and future research into the linked physical
and biological systems of the region.
**2 Southeast Greenland (SEG) study region**
While some fjords, for example Sermilik on the East Coast and Nuup Kangerlua (previously also known as Godthåbsfjord)
on the West Coast, have been studied more extensively, many Greenland fjords have proven difficult to study, including in
Southeast Greenland (SEG). Here, we define the SEG region of interest as extending from 60° N to 70° N (Fig. 1). This





region is of particular interest for a variety of reasons. First, it provides habitat for a genetically distinct polar bear
subpopulation only recently identified (Laidre et al., 2022). Second, it contains particularly remote regions of Greenland
coastline, far from any human settlements and difficult to access for research. Third, it is an area of very high winter
precipitation (Gallagher et al., 2021) and modeling work indicates that it may be one of the last regions of Greenland to
retain substantial coastal land ice (Aschwanden et al., 2019; Bochow et al., 2023). Fourth, it is a region of rapid change, not
only in documented changes to the coastal glaciers and ice sheet (Moon et al., 2020) but also notable declines in offshore sea
ice and warming of coastal ocean currents (Heide-Jørgensen et al., 2022).

## 3 Data and methods

In this study, the fjords in SEG are numbered 1-52 going from north to south (Fig. 1). We also use our own digitized fjord
boundaries created based on synthetic aperture radar (SAR) image mosaics (Cohen et al., 2023; see Code and data availability).
Our analysis is focused on 1 January 2015 through 31 December 2019 to align with SEG polar bear data collection and the
time period of interest established by Laidre et al. (2022).
To characterize a range of environmental metrics, we take advantage of existing data products, such as freshwater flux, iceberg
discharge, and regional climate model output, to create new datasets that support SEG-wide analysis. While remote sensing is
necessary to characterize a region of this scale, the spatial resolution needed (10s to 100s of meters) for some data types is
difficult to achieve from many standard remote sensing products, such as sea-ice cover data products (often with multi-
kilometer resolution). Though researchers are working towards automated classification schemes at the spatial scales needed
for this type of analysis (e.g., Scheick et al., 2019; Soldal et al. 2019), we are unaware of any that can support our specific
study needs. We therefore undertook extensive manual digitization to create landfast sea ice and glacier-derived fjord ice data
records. Along with supporting our analysis, these data (Cohen et al., 2023) should be helpful for ongoing work to improve
machine learning techniques for classifying fjord environments.
Due to the effort required to create manually digitized datasets, we selected eight focus fjords for landfast sea ice and glacier-
derived fjord ice analysis (Fig. 1, Table 1). Our focus fjords include five that were selected for Laidre et al. (2022): Skjoldungen
(63.3º N), Timmiarmiut (62.6º N), Naparsuaq (61.7º N), Anoritoq (61.5º N), and Kangerluluk (61.1º N). These fjords have
been occupied by polar bears for multiple years based on telemetry data collected since 2015 and comprised the core range of
the SEG polar bear population. Here, we expand the fjord selection to include three more northerly focus fjords: Ikertivaq
(65.4° N), Kangerdlugssuaq (68.1° N), and Nansen (68.2° N). Ikertivaq and Kangerdlugssuaq fjords are heavily used by polar
bears that inhabit Northeast Greenland, while their presence was scarcer in Nansen during 2015–2019. The map-view
geometries of our focus fjords (Fig. 1) cover a wide range, from relatively simply shaped long, narrow fjords (e.g., fjords 43
and 48) to complex interconnected channel systems (e.g., fjords 37 and 40).





**Table 1. Focus fjord spatial information, including fjord reference names, areas (km²), and bounding coordinates used for analysis.**

| Fjord Name & Number | Analysis area (km²) | Top Right (lat, lon) | Bottom Left (lat, lon) |
|---|---|---|---|
| Nansen (15) | 375 | (68.43, -29.51) | (68.16, -30.32) |
| Kangerdlugssuaq (18) | 880 | (68.64, -31.52) | (68.05, -32.98) |
| Ikertivaq (31) | 894 | (65.74, -38.96) | (65.36, -40.13) |
| Skjoldungen (37) | 793 | (63.57, -40.80) | (63.08, -41.94) |
| Timmiarmiut (40) | 1079 | (62.98, -41.52) | (62.37, -43.22) |
| Naparsuaq (43) | 182 | (61.83, -42.11) | (61.68, -42.90) |
| Anoritoq (45) | 217 | (61.61, -42.40) | (61.41, -43.12) |
| Kangerluluk (48) | 184 | (61.12, -42.64) | (61.02, -43.64) |

## 3.1 Solid ice discharge across SEG

To compute solid-ice discharge from 2015 through 2019, we used data derived from glacier gates (Mankoff et al., 2020b; Mankoff et al., 2020c). These data were used to create individual glacier discharge time series as well as discharge by-fjord, including daily, monthly, annual and season mean, and cumulative 2015-2019 discharge records (Cohen et al., 2023). Beginning with a glacier dataset evolved from Moon et al. (2020), we manually associated each of these glaciers (shown in Fig. 1) with a glacier gate in the Mankoff et al. (2020b) solid ice discharge dataset; in some cases, there were multiple gates corresponding to a single glacier, and we summed the discharge from these gates accordingly. We filtered out data at times when the dataset coverage attribute was less than 50% (Mankoff et al., 2020b). We also note that some glaciers apparent in satellite imagery are not included in either the Moon et al. (2020) or Mankoff et al. (2020b) datasets (usually because they are narrow and/or slow moving) and are therefore not included in our solid ice discharge results, even though glacier-derived ice in fjords is recorded in a separate dataset (section 3.5).

Solid ice discharge is interpolated for individual glaciers between the first and last dates with observed discharge to create daily time series. We linearly interpolate between observed discharge values to fill data gaps and use the observed discharge and error to calculate the interpolation error (Eqn 15, White, 2017). At the fjord level, the interpolated daily discharge time series for each glacier are summed together, and the fjord discharge error is the root of the sum of the squares of the glacier





discharge errors. The daily time series is then used to construct other solid ice discharge metrics, including a monthly time
series, as displayed in Fig. 9d.
The interpolation procedure, combined with differences in the observational discharge time series length for each glacier,
introduces a small discrepancy between the cumulative discharge from all glaciers and the cumulative discharge from all
fjords (~21 Gt or ~2%). Essentially, the first and last valid dates in the observational time series vary for each glacier, and
interpolation preserves the first and last dates in each discharge time series. Of the 67 glaciers with observed discharge, the
interpolated time series include discharge data starting from 1 January 2015 for 31 glaciers, and ending on or after 26
December 2019 for 65 glaciers. While the glaciers with gaps at the beginning or end of their records were likely discharging,
discharge observations were absent or filtered out for quality, and so the first or last several days in the interpolated time
series for those glaciers are empty. Consequently, of the 33 fjords with observed discharge from at least one glacier, 11
fjords have discharge time series starting from 1 January 2015, and 31 end on 26 December 2019. This results in a slightly
lower cumulative discharge for a fjord than for its component glaciers because fjord discharge is not computed on a date
when any glacier in the fjord has no discharge value. We chose to accept this small discrepancy since it does not impact our
conclusions.
**3.2 Freshwater flux across SEG**
To compute daily time series of freshwater discharge into each fjord from 2015 through 2019, we used freshwater discharge
data, including surface runoff and subglacial discharge, from Greenland land and ice basins (Mankoff, 2020; Mankoff et al.,
2020a). The freshwater discharge data products are created by applying a flow routing algorithm to digital elevation models
of the land and ice sheet surfaces and the ice sheet bed to identify land surface and subglacial streams, stream outlets, and
basins upstream of those outlets. Subsequently, daily runoff from a regional climate model is summed over each of the
identified basins, and instantaneously routed to the appropriate basin outlets. We calculated freshwater discharge into our
fjords by using the command line tool provided with Mankoff et al. (2020a) to identify all outlets within a 500 m buffer of
each fjord boundary; we applied this buffer to account for differences in coastline data products and to ensure that we
captured all freshwater discharge outlets. We then used the command line tool to compute daily freshwater discharge
originating from their predefined land and ice basins and going through the outlets that we identified and into each of our
fjord basins. We used discharge values from both the Modèle Atmosphérique Régional (MAR: Fettweis et al. 2017) and the
Regional Atmospheric Climate Model (RACMO: Noël et al., 2019), both of which were statistically downscaled to a
common 1 km grid and archived for use with these freshwater discharge tools (Mankoff, 2020); we used version 4.2 of the
archival data. Due to a longer time series and to align with other sampled metrics, we relied primarily on the MAR time
series, but we have included the RACMO discharge output in our own archival data.



We also analyzed freshwater discharge variations with depth, including terrestrial runoff and subglacial discharge. We used
the same command line interface and source data (Mankoff et al., 2020a) to identify all freshwater discharge outlets within
each buffered fjord boundaries. These outlet output data include outlet elevation above or below sea level. For outlets above
sea level, we clipped their elevation values to 0 m under the assumption that water flowing from these outlets enters the
fjords at sea level (i.e., surface runoff). Using these data, we calculated daily time series of total freshwater discharge, binned
by discharge depth, for each fjord (for example, Fig. 9c).

**Figure 2. Regions at the mouths of (a) fjords 1-52 and (b) fjords 1-19 (circles of radius 50 km) for offshore sea-ice analysis. Small black dots indicate locations of gridded sea-ice concentration data from AMSR2. Grid cell size is approximately 3.125 × 3.125 km. A buffer zone of three grid cells from land is excluded from analysis due to land contamination of the ocean data, which can be seen in the form of spurious sea ice (red, green, and blue cells) for this date of October 2, 2013, when sea ice is almost surely not present along this portion of the coast. The black circles are associated with the focus fjords of this study.**



**3.3 Sea ice and sea surface temperature**

To characterize the offshore sea ice at the mouths of the fjords, we used sea-ice concentration data derived from the passive microwave AMSR2 (Advanced Microwave Scanning Radiometer 2) instrument onboard the GCOM-W satellite operated by the Japan Aerospace Exploration Agency (Kaleschke and Tian-Kunze, 2016). The brightness temperature data were processed at the University of Hamburg using the ASI algorithm (Beitsch et al., 2014) to create daily gridded fields of sea-ice concentration with nominal grid cell size $3.125 \times 3.125$ km. We defined circles of radius 50 km centered at the mouths of the fjords (Fig. 2a). Within each circle we identified the offshore grid cells, excluding a buffer zone of three grid cells from land because the sea-ice signal in those cells may be contaminated by the signal from land (Fig. 2b). We then calculated the daily sea-ice area for the valid grid cells within each circle during 2015-2019. Figure 9a shows an example, in which the black curve is the daily sea-ice area, and the purple curve is a 31-day running mean. We defined a threshold equal to 15% of the mean March-April sea-ice area (horizontal black dotted line) and found the dates each year when the 31-day running mean crossed the threshold (vertical yellow dashed lines). The date in the spring when the sea-ice area drops below the threshold on its way to the summer minimum is called the spring transition date; the date in the fall when the sea-ice area climbs above the threshold on its way to the winter maximum is called the fall transition date. The transition dates for all fjords and all years are shown in Fig. 6.

To include further comparison metrics for sea ice coverage and also sea surface temperatures at the fjord mouth, we sampled output from MARv3.12 (Fettweis et al. 2017). MAR results have a grid resolution of 6.5 km, and we sample a single grid cell centered at the fjord mouth, which we extract based on fjord mouth outlines created as a subset of developing the SEG fjord boundaries (e.g., Fig. 1; Cohen et al., 2023). The FRA variable identifies the open water and sea ice cover percentages, while the ST2 variable provides the sea surface temperature (SST) for open water and sea ice surface temperature. These are used together to determine the percent sea ice cover and the SST for the open water fraction. MAR has a hard-coded maximum sea ice cover of 95%, which we retain in our plotted results (e.g., Fig. 9e). Note that MAR assimilates SST and sea ice cover data from ERA5 available at a resolution of 0.3 x 0.3° (Hersbach et al. 2020).

**3.4 Landfast sea ice for 8 focus fjords**

To analyze landfast sea ice, we combined data extracted from imagery via the Operational Land Imager (OLI) onboard the USGS Landsat 8 satellite with data extracted from images captured by the Moderate Resolution Imaging Spectroradiometer (MODIS) instruments aboard the NASA Aqua and Terra satellites. There are notable differences between the two datasets: Landsat 8 imagery provides higher spatial resolution (30 m) with lower temporal resolution (16-day repeat cycle for each image footprint), while MODIS has lower spatial resolution (250 m) but higher temporal resolution (daily). Clouds and polar night limit the functional temporal resolution of both Landsat 8 and MODIS as the two satellites operate using optical sensors..



The suitability of every image from 1 January 2015 through 31 December 2019 in the region of interest was manually
inspected for use in our analysis. MODIS imagery was obtained from the NASA Worldview website
(https://worldview.earthdata.nasa.gov) and we downloaded the Corrected Reflectance (True Color) images that were
determined to be cloud-free (Fig. 3a). We used the USGS EarthExplorer web tool (https://earthexplorer.usgs.gov) to preview
all available Landsat 8 imagery and evaluate cloud cover (with a starting filter of 90% cloud cover). We downloaded cloud-
free Collection 1, Level 1 data (Fig. 3a) and we created multi-band natural color images using bands 4, 3, and 2. We used
both the R "stack" tool included in the "raster" package (https://cran.r-project.org/web/packages/raster/raster.pdf) and the
Composite Bands (Data Management) tool in ArcGIS to produce these composites. These composite imagery datasets were
catalogued and served as the foundation for further analysis.
Glacial ice, landfast ice, and pack ice share similar visual characteristics and are often adjacent to or intermixed with one
another within SEG fjords. Larger fjord systems, where active glaciers introduce glacial ice and large fjord mouths facilitate
the accretion of pack ice inside the fjords during the frozen season, are especially likely to contain a mixture of ice types.
This is compounded by the intricate geometry of these fjord systems, in which narrow corridors or tortuous coastlines entrap
ice of various types. Thus, we worked to distinguish landfast ice from glacier-derived ice, open water, and pack ice floes
(Fig. 4). By having one person complete the entirety of the digitization process, we attempted to reduce the potential
sensitivity of our manual analysis procedure.





**Figure 3. Data availability during 2015-2019 for a) fast ice analysis from MODIS and Landsat 8 images covering day 0-180 and b) glacial ice analysis from Landsat 8 images covering the full year.**

Several visible characteristics in Landsat 8 imagery facilitated the identification of landfast ice: a smooth surface texture
(especially relative to glacier-derived ice); bright surface character; image-to-image persistence; and adhesion to coastal
boundaries. Landfast ice is more challenging to distinguish in lower-resolution MODIS imagery. Regarding identification of
landfast ice in MODIS images, pixel color was the most useful identifier along with image-to-image persistence. Several
smaller regions in our study area were poorly resolved by MODIS imagery, resulting in varying optical properties (e.g.,
color, saturation, brightness) for otherwise consistent ice surface characteristics. To address this issue, the higher-resolution
Landsat 8 imagery was analyzed first and produced landfast-ice boundaries with a higher level of accuracy on the dates when



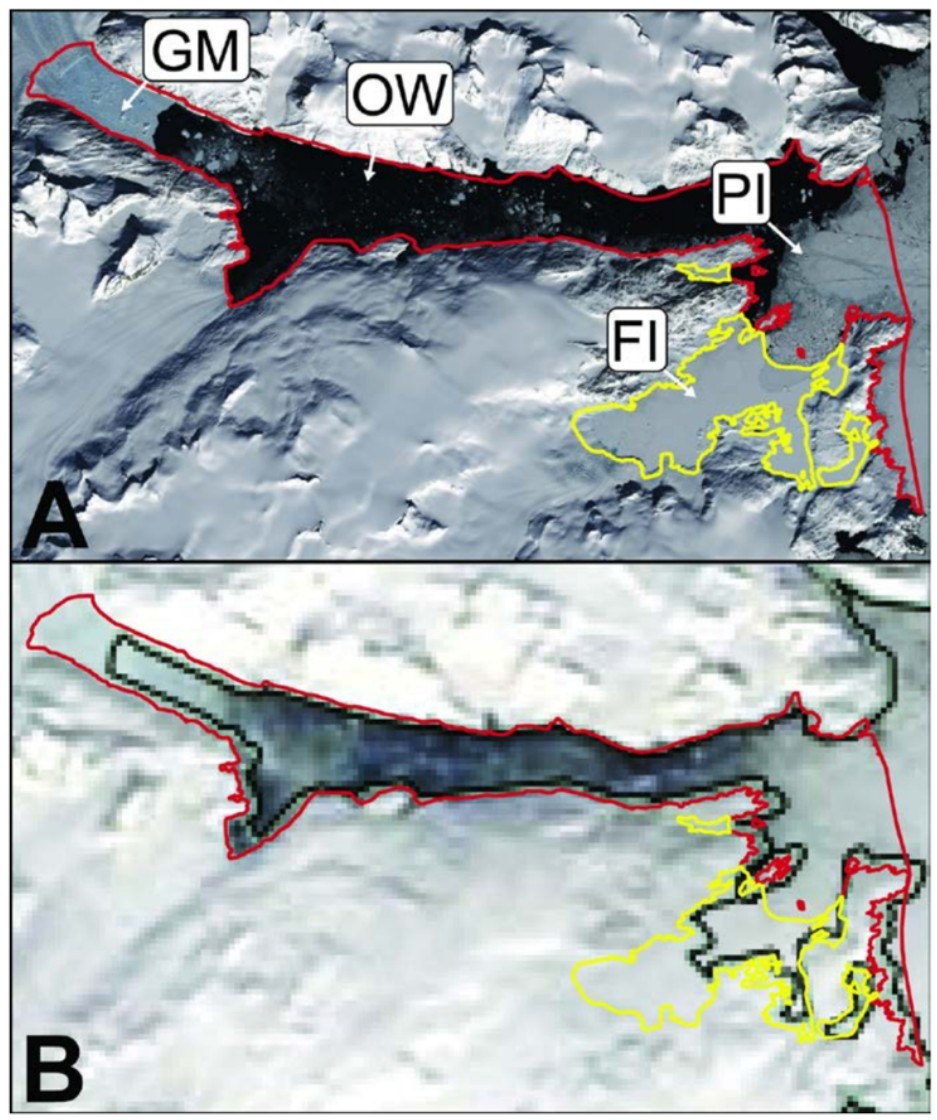

**Figure 4. Example fast ice digitization. (a) Landsat 8 and (b) MODIS image examples for Anoritoq Fjord, both from 7 April 2017. Yellow outlines identify the fast ice areas and red lines indicate the rest of the fjord boundary. Note the distinct visual character of glacial mélange (GM), open water (OW), fast ice (FI), and pack ice (PI) (indicated in a). The misplacement of the coastline in the standard MODIS product is also apparent (b), and we use our own fjord boundary product for analysis. Figure reproduced from Laidre et al. (2022).**

such images were available. The MODIS imagery was processed afterwards, using the results of the Landsat 8 analysis as a
guide for the characterization of MODIS imagery. This facilitated increased accuracy of digitization within areas of
ambiguous interpretation (as described below).



To quantify the degree of error introduced by using MODIS when Landsat 8 was unavailable, we digitized 25 MODIS
images (analyzing 1 image from 2015-2019 for Skjoldungen, Timmiarmiut, Naparsuaq, Anoritoq, and Kangerluluk fjords)
captured on the same date as Landsat 8 images already analyzed. We found a mean difference between the results of MODIS
and Landsat 8 digitization of 1.2 $km^2$ of fast-ice area and a standard deviation of 12.6 $km^2$. These levels of disagreement
have no significant impact on our conclusions.
Based on early results, landfast sea ice boundaries were analyzed starting on January 1 until either July 1 or ice-free
conditions were reached (whichever was first) from 2015 through 2019. We manually delineated landfast-ice boundaries for
each available image. Based on visual analysis, we traced landfast-ice boundaries (without regard to fjord edge boundary)
and recorded the date and source of the image. Any portions of the resulting polygons outside of the fjord boundaries were
erased using the Clip (Analysis) tool in ArcGIS, which resulted in fjord-surface measurements of landfast-ice area and
percent area coverage. This method precluded repetitive and time-consuming fjord boundary tracing, allowing for rapid
digitization of landfast ice.
After calculating the landfast-ice area in a fjord system from all available imagery within a single year, we applied a moving
average to obtain a smooth representation of the formation and breakup of landfast ice. The moving average on day $t$ is
calculated using weights proportional to $\exp(-\Delta t^2/T^2)$ where $\Delta t$ is the number of days from $t$ to other data points, and $T$ is a
time scale equal to 7 days. To demonstrate the likelihood of landfast ice presence in any given spatial region across all
observations, we also produced "heatmaps" of landfast sea ice presence (Figs. 10-13a,c) by overlaying all individual spatial
occurrence maps and applying a gradient of shading (applying grid cell size of 50 m x 50 m).
**3.5 Glacier-derived ice for 8 focus fjords**
To analyze glacier-derived ice, we again used USGS Landsat 8 data imagery (following section 3.4 methods). The low
spatial resolution of MODIS imagery made it unsuitable for this analysis. Because glacial ice has a year-round presence, we
analyzed glacial ice presence from 1 January 2015 to 31 December 2019 for each year (Fig. 3b).
We characterized glacier-derived ice using four primary categories (Fig. 5, Table 2): spatially dense glacial ice mélange
(type 3); moderately high-spatial-density, mixed-size glacier-derived ice with large icebergs (type 2); low-spatial-density
glacier-derived ice with large icebergs (type 1); and consistent small-ice surface without large icebergs (type 0). (We also
used a 'type 99' classification for glacier ice not yet calved). To measure the temporal and spatial distribution of glacier-
derived ice in SEG, we analyzed the optical satellite imagery from Landsat 8 using the same ArcGIS 10.8 method as
described for landfast sea ice for each glacier-derived ice type (Table 2). For the heatmaps of glacial ice presence (Figs. 10-
13b,d), we combine spatial extent for type 2 and type 3 glacier-derived ice. This is motivated by an assessment that type 2
and type 3 glacier-derived surface ice is more feasible for use as polar bear habitat platforms (e.g., Laidre et al., 2022).



**Figure 5. Example glacial ice digitization for Anoritoq fjord (fjord #45). Landsat 8 (8/1/15) background image showing the fjord boundary (red outline) and the digitized zones of different glacier-derived ice types on the fjord surface (green outlines and type indicated): type 3 (dense glacial mélange), type 2 (mixed glacier-derived ice), type 1 (small glacier-derived ice), type 0 (highly dispersed glacier-derived ice), and type 99 (glacier surface) (see Table 2). The boundaries are combined to determine final values for glacier-derived ice area.**

## 4 Results

This study includes data sets that span Southeast Greenland and metrics assessed only for the eight focus fjords. This supports some SEG region-wide analysis and further analysis to include more ocean-surface ice metrics for the eight focus fjords. Along with providing a more complete picture of the SEG environment, these results can support ongoing research into the current and future biological uses of SEG coastal fjords.



**Table 2: Glacier-derived fjord ice types as applied in this analysis.**

| Glacial Ice Type | Description Used for Manual Digitizing |
|---|---|
| Type 3 (dense glacial mélange) | White to pale to blue color. Color (considering variation in texture) consistent throughout with bright, vibrant character<br>Appears potentially cohesive, without open water gaps. May have sharp edge boundaries<br>Texture: clear inclusions of many icebergs<br>Also digitize very large (~>1km width) mélange platforms |
| Type 2 (mixed glacier-derived ice) | Majority of ice colored grayish blue of varying shades with semi-transparent character<br>Discernible floes of apparently glacial origin, varying size with inconsistent cohesion and potential presence of small (~<250 m) open water gaps. Possible presence of Type 3 platforms<br>Includes sizable icebergs |
| Type 1 (small glacier-derived ice) | Gray blue to dark blue coloration with higher degree of transparency compared to Type 2 and Type 3 ice<br>Little to no cohesion, but still high spatial concentration of likely growlers/bergy bits. Few icebergs and Type 3 platforms of any substantial size, but not absent |
| Type 0 (highly dispersed glacier-derived ice) | Concentration of icebergs of moderate size (~250 m width) > 10% and <30%<br>Little slushy (grey) background ice (bergy bits, growlers) |
| Type 99 (glacier surface) | Glacier surface. Sections of glacier ice not yet calved but inside the fjord boundary. |

**4.1 Regional-scale observations**

Datasets for offshore sea ice, freshwater flux, and solid ice discharge support an examination of conditions across the full SEG region of interest.

**4.1.1 Offshore sea ice**

Figure 6 shows the spring and fall transition dates for offshore sea ice at each fjord. First, while there is substantial year-to-year variability in the spring transition dates, which range from May to early August, there is little variability with latitude for a given year. In other words, offshore sea ice tends to disappear from the coast of SE Greenland in spring over a relatively short time interval across all latitudes, but the timing of that disappearance varies from year to year. Second, the arrival of offshore sea ice in the fall has a narrower range of interannual variability, but there is a distinct dependence on latitude, with sea ice arriving in October at the more northerly fjords and in January or early February at the more southerly fjords. The different nature of the spring and fall transition dates may be due to the relative influence of thermodynamics vs. dynamics. In spring, rising temperatures along the coast may melt the sea ice at more-or-less the same time at all latitudes. But in fall, the arrival of sea ice is due to transport from the north (via the East Greenland Coastal Current) rather than freezing in place. A sea-ice "front" progresses from north to south every fall, at a speed of roughly 10 km day$^{-1}$ (Fig. 6). Note that previous research identified that sea ice along the SEG coast had a mean wintertime (January-April) south-moving speed



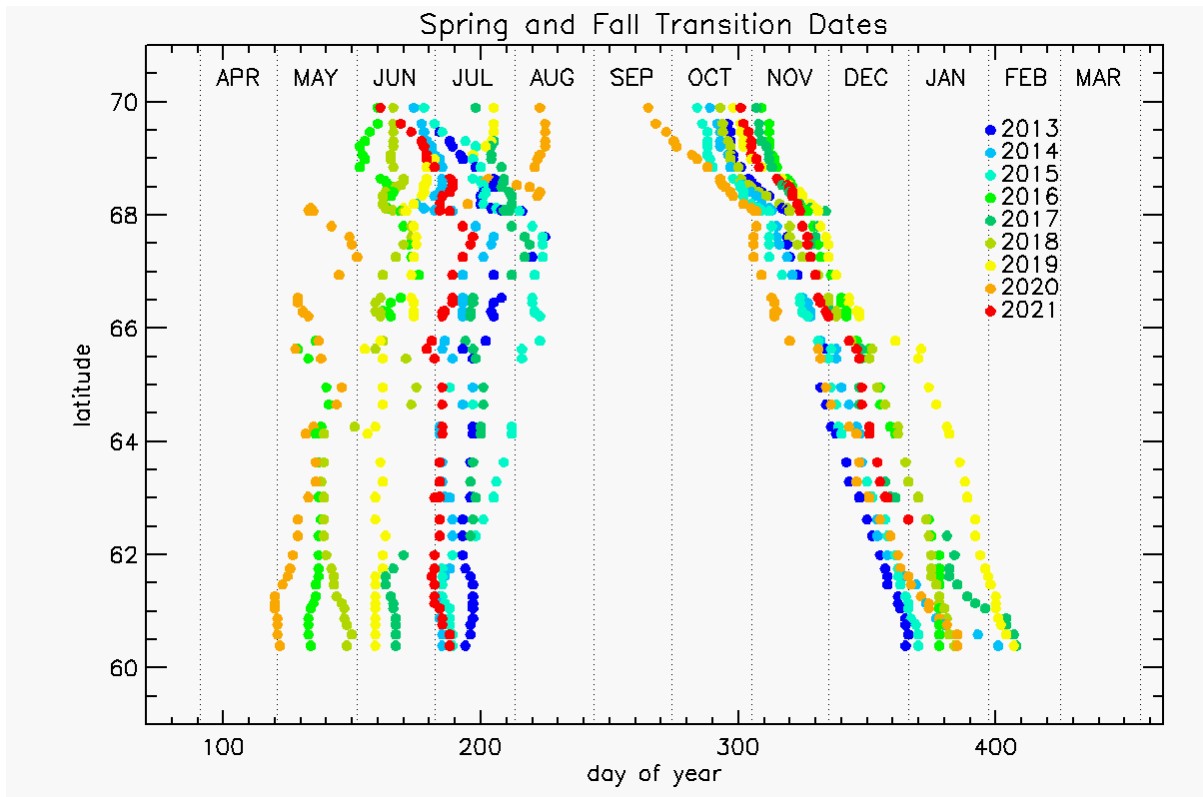

**Figure 6. Spring and fall transition dates of offshore sea ice for all fjords (by latitude) and years (by color) based on a 15% coverage threshold.**

of about 15 cm s$^{-1}$ (13 km day$^{-1}$) from 2010 to 2018 (Laidre et al., 2022). In spring, the sea ice does not retreat along a well-
defined front. Though the seasonal coverage and concentration of offshore sea ice during our study period is reduced from
earlier decades (Heide-Jørgensen et al. 2022) and is expected to continue to shorten and decline, respectively (Kim et al.,
2023), we suggest that the differences in spring and fall transitions may largely persist (while sea ice is still forming).
**4.1.2 Freshwater flux**
Figure 7 shows freshwater flux on the fjord scale across SEG. The results show that there is large variability, from low total
annual discharge of ~1x10$^8$ m$^3$ (~0.1 Gt) at fjords 6 and 44 up to ~1.25x10$^{10}$ m$^3$ (~12.5 Gt) at Sermilik Fjord (fjord 30),
though notably the next largest fjord freshwater fluxes are only 8.48x10$^9$ m$^3$ (8.48 Gt; Kangerdlugssuaq, fjord 18) and
7.12x10$^9$ m$^3$ (7.12 Gt; Jens Munk, fjord 33). In the northern region of SEG, the catchment geography feeds much of the
freshwater to fjord 5, while other fjords in that zone see little freshwater flux until reaching south to fjord 15 and then to
fjord 18 (Kangerdlugssuaq). There's low to moderate flux for most fjords between 18 and 30 (Sermilik), with a notable
increase in mean annual freshwater flux for a number of fjords south of Sermilik.

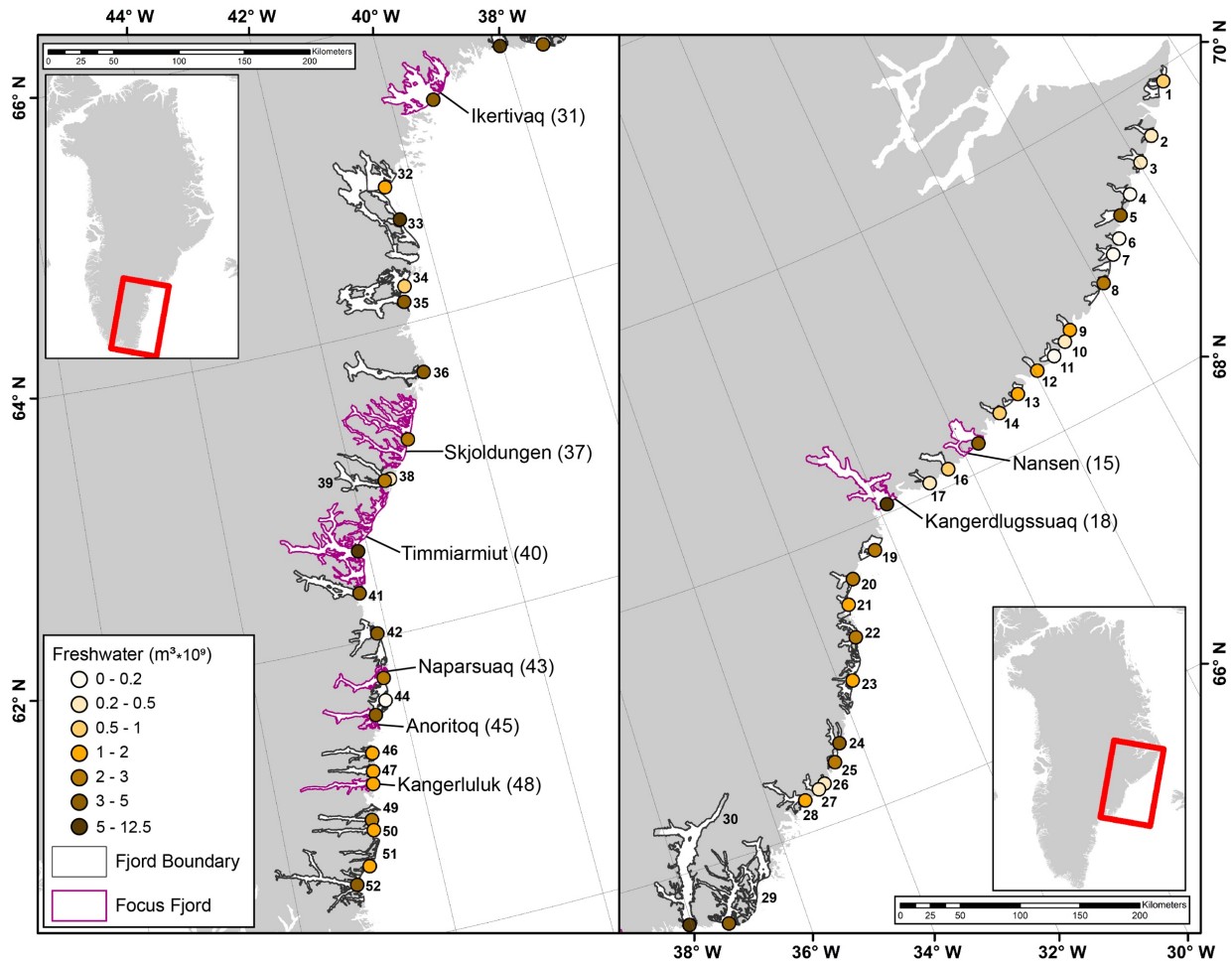

**Figure 7. Mean total annual freshwater flux (m³x10⁹) for 2015 through 2019. The freshwater discharge is summed for the full fjord, including melt that originated from ice-covered and terrestrial areas and sourced from Mankoff (2020) and Mankoff et al. (2020a). Note that for freshwater, 1 m³x10⁹ volume is equivalent to 1 Gt weight.**

Using the discharge elevation/depth, we were also able to assess how much freshwater was entering fjords at the ocean
surface or at depth, discharging from under marine-terminating glaciers. Across the SEG study region, the ocean surface
input and 0-20 m depth bins receive the most input when considering flux through sea level to 1000 m depth (Fig. A1).
Across the region and looking deeper into the water column, flux totals are highest within the top 100 m. While flux is
measured as deep as 900 m (fjord 31, Ikertivaq), most flux occurs at depths shallower than 600m. Strong seasonal variability
in freshwater flux is also apparent (e.g., Fig. 9c). Detailed individual fjord plots are available via our research code (see Code
and data availability).



## 4.1.3 Solid ice discharge



**Figure 8. Mean annual solid ice discharge (Gt/yr) during 2015 through 2019 for glacier-derived ice from indicated glaciers, calculated using Mankoff et al. (2020b).**

Figure 8 shows annual solid ice discharge estimates. We used a fjord-scale perspective to examine solid ice discharge and relied on the availability of glacier solid ice discharge data from Mankoff et al. (2020b, 2020c). Because of this, our solid ice discharge values may underestimate discharge or provide no data for a fjord in which some glacier-derived ice is variably present. For example, the source dataset contains no glacier discharge data for Skjoldungen fjord even though glacier-ice inputs are apparent in our satellite image analysis (Figs. A4 and 11d). Within the fjord dataset we were able to create (Fig. 8), fjords north of Sermilik have relatively small annual contributions of glacier-derived ice, with the exception of Kangerdlugssuaq (fjord 18) and, to a lesser extent, fjord 21. Slow flow rates and often relatively thin glacier termini in this region are the cause of the low glacier-derived concentrations in many fjords, especially for the Geike Plateau, where most



**Figure 9: Time series for fjord 15 (Nansen) showing: a) daily (black line) sea-ice area (km²) and percent coverage based on AMSR-2 sea ice concentration, along with a 31-day running mean (purple), b) area (km²) and percent coverage for landfast ice evaluated from MODIS (blue dot) and Landsat (purple dot) single image sources and with smoothed (blue) record and for all four surface character types (0-3) for glacier-derived ice, c) total freshwater flux (m³ s⁻¹, black dashed line) and depth-binned (solid line) freshwater flux, d) cumulative fjord solid ice discharge (Gt yr⁻¹), and e) sea surface temperature (black line) and sea ice coverage (purple line) measured at the fjord mouth from MAR climate data. Vertical dashed orange lines in all panels indicate the freeze-up and break-up dates for offshore sea ice (panel a) as measured by passing a threshold of 15% of mean March-April sea ice area. A similar threshold is indicated (dashed line) in panel e, while panel b is a simple 15% threshold (dashed line). Similar figures are provided in Appendix A for other focus fjords.**

glaciers may be considered part of a peripheral ice cap (Rastner et al., 2012). By contrast, Ikertivaq and a number of fjords



south of Sermilik are fed by several glaciers, many of which receive moderate and greater levels of solid ice discharge.
**4.2 Focus fjord observations**
Manual analysis of landfast sea ice and glacier-derived ice allows us to integrate these observations and compare across
metrics. Figs. 9 and A2-8 provided stacked 2015-2019 time series of offshore sea ice area and percent coverage; landfast ice
and glacier-derived ice area and percent coverage; freshwater flux binned into sea surface input and input at depths of 0-100
m, 100-200 m, and >200 m; cumulative fjord solid ice discharge; and fjord mouth SST and sea ice coverage from
MARv3.12. These give a sense of temporal evolution across a range of latitudes. In contrast, Figs. 10-13 hone in on results

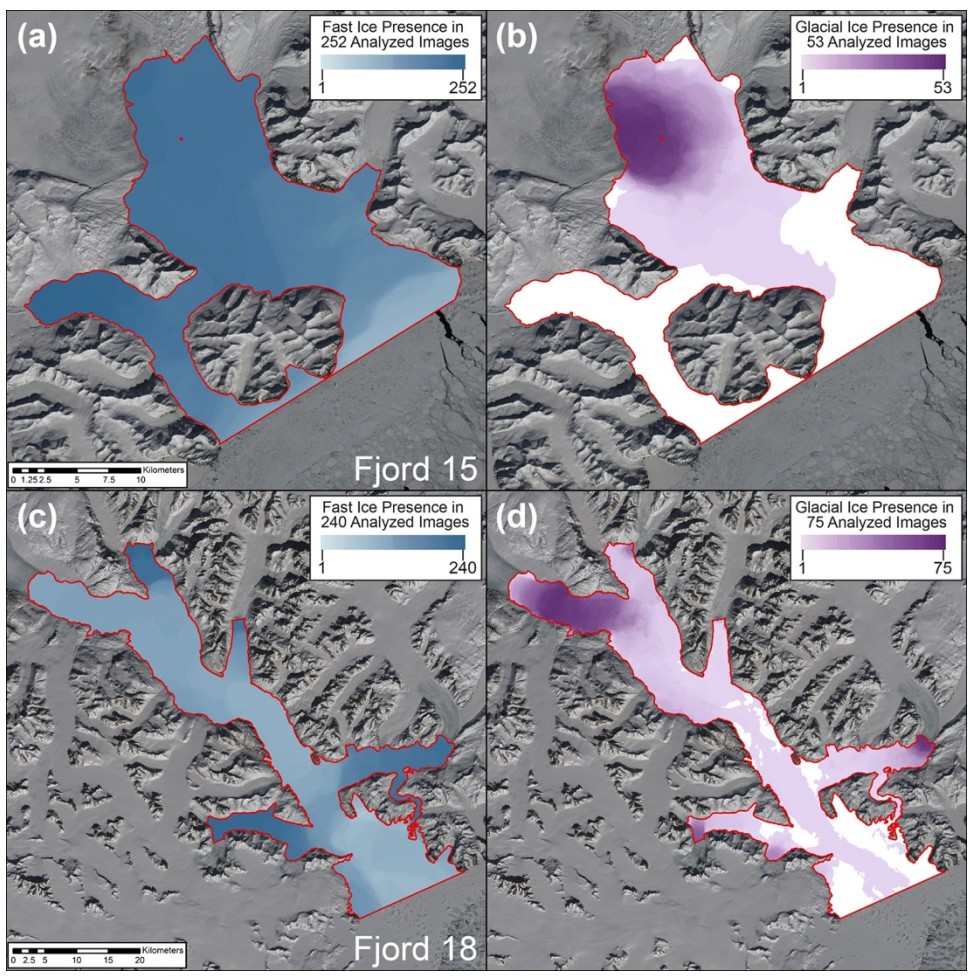

**Figure 10. Maps of fast ice presence (a and c) and glacial ice presence for types 2 and 3 (b and d) for fjord 15 (Nansen, top panels) and fjord 18 (Kangerdlugssuaq, bottom panels). Map symbology is relative to the number of images analyzed (noted in panel legends).**




of the landfast and glacier-derived ice analysis to provide a spatial map-view for the presence of landfast ice and types 2 and
3 glacier-derived ice.
Across all eight focus fjords, landfast ice regularly accumulates in particularly narrow fjord "corridors" (narrow areas of the
fjord with entrances/exits for ice flux on either end; e.g., Fig. 11a, c) and/or the "corners" of fjords (areas with a single
entrance/exit for ice flux and a confined coastal topography; e.g., Fig. 12a, c). The Nansen (fjord 15) and Kangerdlugssuaq
(fjord 18) fjords display periods in which they are fully covered by landfast ice in certain years, while all the more southerly
fjords do not reach full landfast ice coverage in any study years.
Despite broad seasonality and spatial consistency to landfast ice development, there is substantial year-to-year variability for
landfast ice development within each fjord (panel b within Figs. 9 and A2-8). When considering a 15% landfast ice coverage

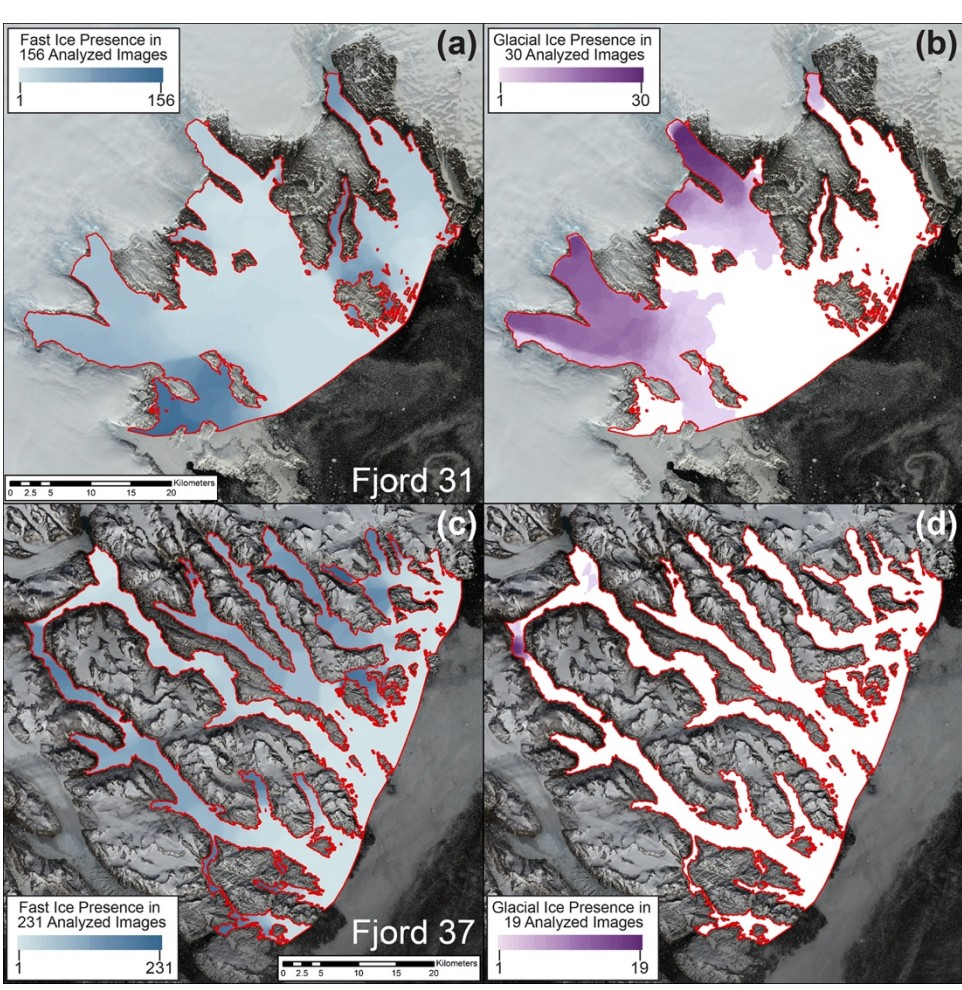

**Figure 11. Same as figure 10 for fjord 31 (Ikertivaq, top panels) and fjord 37 (Skjoldungen, bottom panels).**



threshold, more northern five focus fjords have lower variability in the timing of landfast ice development and breakup, but

the timing of the fast-ice peaks have substantial variability (Table A1). For example, in 2017 in Ikertivaq the landfast ice was

slower to form, with some expansion/decline, before peaking at close to 80% area coverage in late April, while in 2019

Ikertivaq experienced a relatively rapid development of landfast ice with a similar area coverage peak in early March (Fig.

A3). For the three southernmost fjords there is larger variability in the timing of the formation and breakup of the landfast

ice. Landfast ice did not surpass a 15% ice coverage threshold for Naparsuaq in 2019, Anoritoq in 2015, and Kangerluluk in

both 2015 and 2019 (Figs. A6-8). Yet, we do observe clear instances of landfast ice remaining in place well-after offshore

sea ice has fully disappeared, with many of the focus fjord declines in landfast sea ice lagging the offshore sea-ice declines

by more than a month in 2016 and ~two weeks in 2018 (panel b within Figs. 9, A2-8).

Glacier-derived ice presence for types 2 and 3 combined (Figs. 10-13b,d) is dependent on marine-terminating glacier

locations, with higher presence near the glacier termini. As expected, the manually digitized imagery also highlights glacier

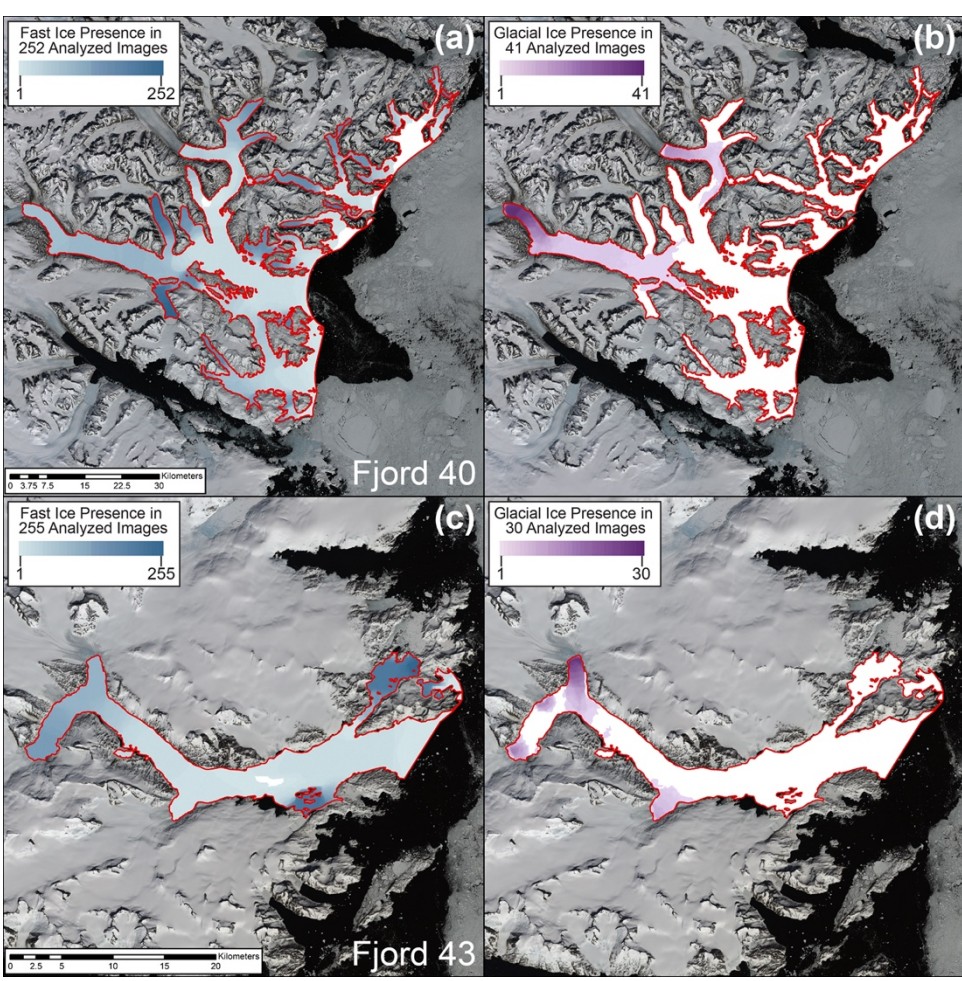

**Figure 12. Same as figure 10 for fjord 40 (Timmiarmiut, top panels) and fjord 43 (Naparsuaq, bottom panels).**



ice inputs that may be absent in other datasets (such as we use for regional SEG solid ice discharge). Because of landfast ice
and glacier-derived ice intermixing (or at minimum an inability to distinguish boundaries from satellite imagery), our results
highlight glacier-derived ice-dominant or landfast ice-dominant fjord regions rather than consistent or clear delineations
within most fjord regions. The time series of glacier-derived ice (Figs. 9, A2-8) indicate that only Kangerdlugssuaq,
Ikertivaq, and Anoritoq more regularly contain types 2 and 3 glacier-derived ice outside of that fjord's landfast ice season.

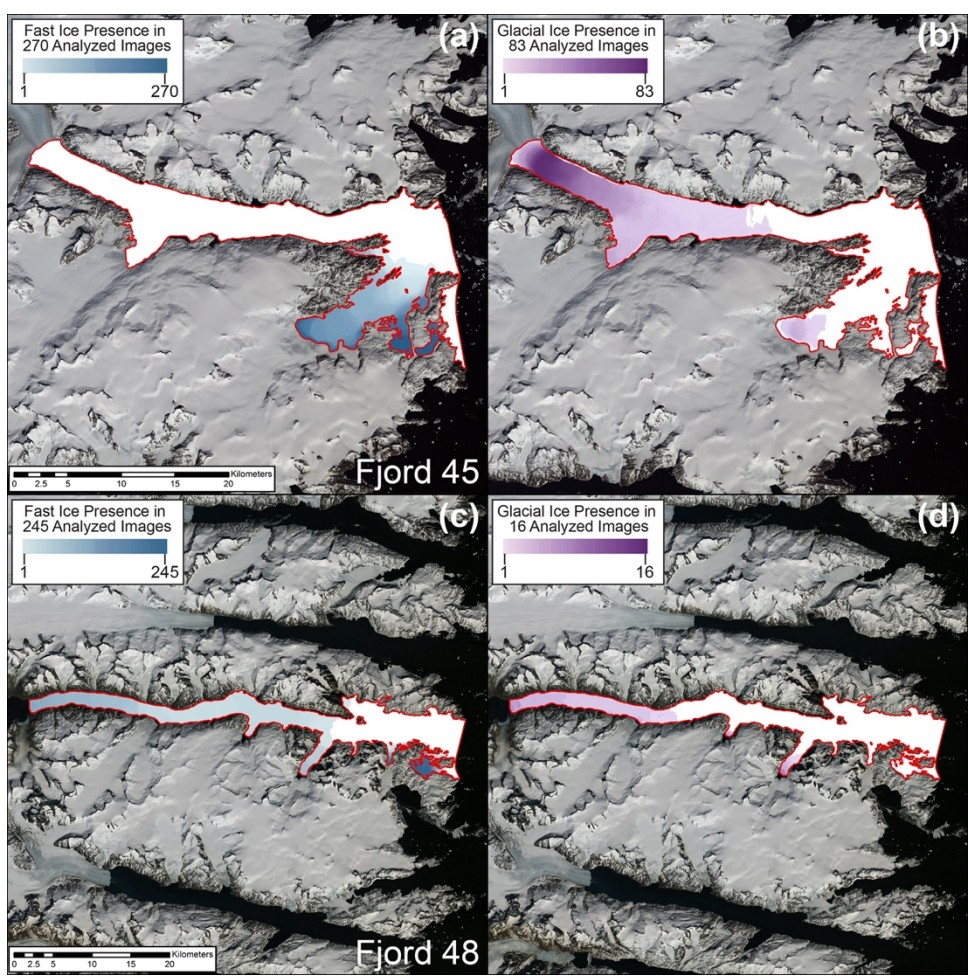

**Figure 13. Same as figure 10 for fjord 45 (Anortioq, top panels) and fjord 48 (Kangerluluk, bottom panels).**

## 4 Discussion

Factors affecting ice in SEG fjords can be broadly divided into two categories: (1) fixed factors such as fjord width, length,
bathymetry, orientation, latitude, and locations of glaciers feeding into the fjord; and (2) variable factors such as katabatic
winds coming off the ice sheet, along-shore winds driven by cyclones, ocean currents, ocean stratification, ocean heat



content, air temperature, formation of sea ice, and the discharge of freshwater and glacial ice into the fjord. The formation of
landfast ice and accumulation of glacier-derived ice in SEG fjords tends to have a semi-consistent spatial pattern; landfast ice
and glacial ice can be found in similar areas within each individual fjord from year to year (Figs. 10-13). This distribution is
likely a combination of fixed and variable factors. For example, the morphology of each fjord system is likely a first order
control. Variable factors such as ocean currents may also produce relatively consistent ice conditions, but current and future
potential for ocean variations have to be considered. For example, as the East Greenland Coastal Current flows past the
mouth of a fjord, it turns to the right (due to Coriolis) and enters the fjord, keeping the shoreline on the right. The current
flows into the fjord along the north or east side of the fjord, then out along the south or west side of the fjord, influencing
ice-forming surface conditions and iceberg motion in the process. But the flow is not steady in time. Recent examination of
four East Greenland fjords, including two in SEG (Kangerdlugssuaq and Sermilik), found periodicity in current patterns in
the range of 2-4 days for Kangerdlugssuaq, plus a broad peak around 10 days (Gelderloos et al., 2022). Thus, factors still not
included in this study warrant examination and future synthesis.
Temporally, landfast ice and glacial ice follow different patterns. Landfast ice forms seasonally from roughly February to
late May, with significant inter-annual variability of cover duration (Table A1), while glacier-derived ice can be found in
various fjords year-round (Laidre et al., 2022). However, the character (e.g., type 0-3), timing, and area coverage of glacier-
derived ice is strongly fjord-dependent, with even some glacier-fed fjords appearing to provide little possibility for
substantial glacier-derived ice habitat outside of the landfast ice season.
Of note regarding our mapping of landfast ice locations is that they commonly appear in areas that remain poorly mapped for
bathymetry. Comparing landfast ice locations with bathymetric data from BedMachine 5 (Morlighem et al., 2017;
Morlighem et al., 2022), for example, landfast ice often occurs in presumably shallow regions that lack any bathymetric
detail. Greenland sea level responses to climate change include the possibility for local regions to experience falling sea
levels (Fox-Kemper et al., 2021). This suggests that understanding shallow-region bathymetry will only become more
important, though the sea level changes may occur much slower than some other global coasts. For example, changes in
ocean depth have the potential to influence wave character, which contributes to mechanical landfast ice breakup (Petrich et
al. 2012), and the prevalence of possible grounding points, which may influence landfast ice formation (Mahoney et al.

345  2014).

Glacier-derived ice, produced from marine-terminating glaciers in SEG fjords, is initially deposited at the glacier terminus
and proceeds to drift into the fjord as it melts, fractures, and disperses. As glacial ice travels through the fjord system, it can
become trapped amongst forming landfast ice and thus effectively adding to the landfast ice itself. This is especially frequent
in narrow, long fjords where landfast ice can clog passageways and prevent glacial ice from exiting the fjord at the mouth.
This heterogeneous mixture of frozen fast ice and glacial ice provides stable optimal springtime habitat for ice-breeding
seals, as well as foraging polar bears (Laidre et al., 2022). The distribution of glaciers across SEG (e.g., Fig. 1) is
heterogeneous, with some fjord systems having multiple productive glaciers (e.g., fjords 18 and 31) while others have minor





or no glacier-derived flux (e.g., fjord 37). It is unclear from our observations the extent to which glacier-derived ice either
enhances landfast ice persistence or diminishes it. For example, production of glacial ice in fjord 15 may help to compress
and possibly thicken landfast ice (Fig. 10a,b), especially if paired with sea ice circulating into the fjord from offshore. On the
other hand, glacial ice traversing from a glacier terminus towards the fjord mouth might shear against the landfast ice edges,
particularly if they are subject to different wind or current forces, for example due to different surface heights and bottom-ice
depths.
Differences in offshore sea ice and landfast ice development across SEG suggest that glacier-derived ice may be especially
important as a fjord surface ice environment. Earlier research demonstrated that the 1999-2018 mean width of the wintertime
sea-ice band for 60-65°N was 19 km, while for 65-70°N it was 149 km (Laidre et al. 2022). The four most southerly focus
fjords functionally experienced no full coverage of offshore sea ice throughout 2015-2019 (Figs. 9a, A2-8a). Combined with
low landfast ice coverage, animals may have limited options for sea ice platforms, while glacier-derived ice is present to
some extent in all of these fjords. The extent to which limited and sporadic coverage of glacier-derived ice (Figs. 9b and A2-
8b) provides year-round ice habitat is unknown, but observations and tracking data of top predators suggests animals use this
habitat year-round for hauling out (e.g., resting) or foraging (Laidre et al., 2022).
**5 Conclusion**
Fjords across Southeast Greenland exhibit high fjord-to-fjord variability in regards to bathymetry, size, shape, and glacial
setting. As a result, some fjords receive substantially higher annual freshwater flux from ice sheet/glacier and terrestrial
runoff, as well as fjords with much higher presence of glacier-derived ice. The inputs mix with in-fjord sea ice and landfast
ice and offshore sea ice to create a dynamic fjord surface environment.
Across 2015 through 2019, SEG fjords demonstrate substantial year-to-year variability. While the impacts of climate change
may be expected to push long-term trends in one general direction, the variability in separate metrics will likely be different.
For example, the sensitivity of freshwater flux to ice sheet surface melt introduces a high dependency on atmospheric
conditions, which change rapidly and have high inter-annual variability (Lenaerts et al., 2019). On the other hand, solid ice
discharge depends on ice sheet and glacier dynamics, which generally respond more slowly to climate change and have
lower inter-annual variability (Moon et al., 2022), and ocean conditions. Landfast sea ice variability introduces further
dependence on ocean surface conditions, which are also a major factor for formation of mobile sea ice.
With sea ice loss well underway along the SE Greenland coast and projections for summer sea-ice free conditions to occur
within one to two decades (Kim et al. 2023), the importance of glacier-derived ice as a habitat for top predators may only
rise. Projections for the spatial patterns of Greenland Ice Sheet retreat under a range of future scenarios point towards the
longer-term presence of glacier ice in SEG compared to other areas on the coast (Aschwanden et al., 2019; Bochow et al.,
2023). High winter precipitation in SEG as compared to other regions (Gallagher et al., 2021) is one important factor in



sustaining glacier ice in the region. This higher regional winter snowfall may also provide longer-term habitat appropriate for
ringed seal birthing lairs, which are created as on-sea-ice snow caves with sufficient snow cover associated with lower
predation rates (Kelly et al., 2010). Further, the heterogeneous mix of glacial ice frozen into the fast ice can provide suitable
drifts for ice seal birth lairs, which can form quickly on any side of an iceberg given their complex geometry. This has also
been seen in the case of polar bear maternity dens in Northeast Greenland (Laidre and Stirling 2020). As a result, there is a
potential for SEG to remain a long-term (century to millenia scale, dependent on future climate change pathway) refugia
location for polar bears and other ice-dependent wildlife, but further investigation is required to quantitively assess this
potential.
**Appendix A**

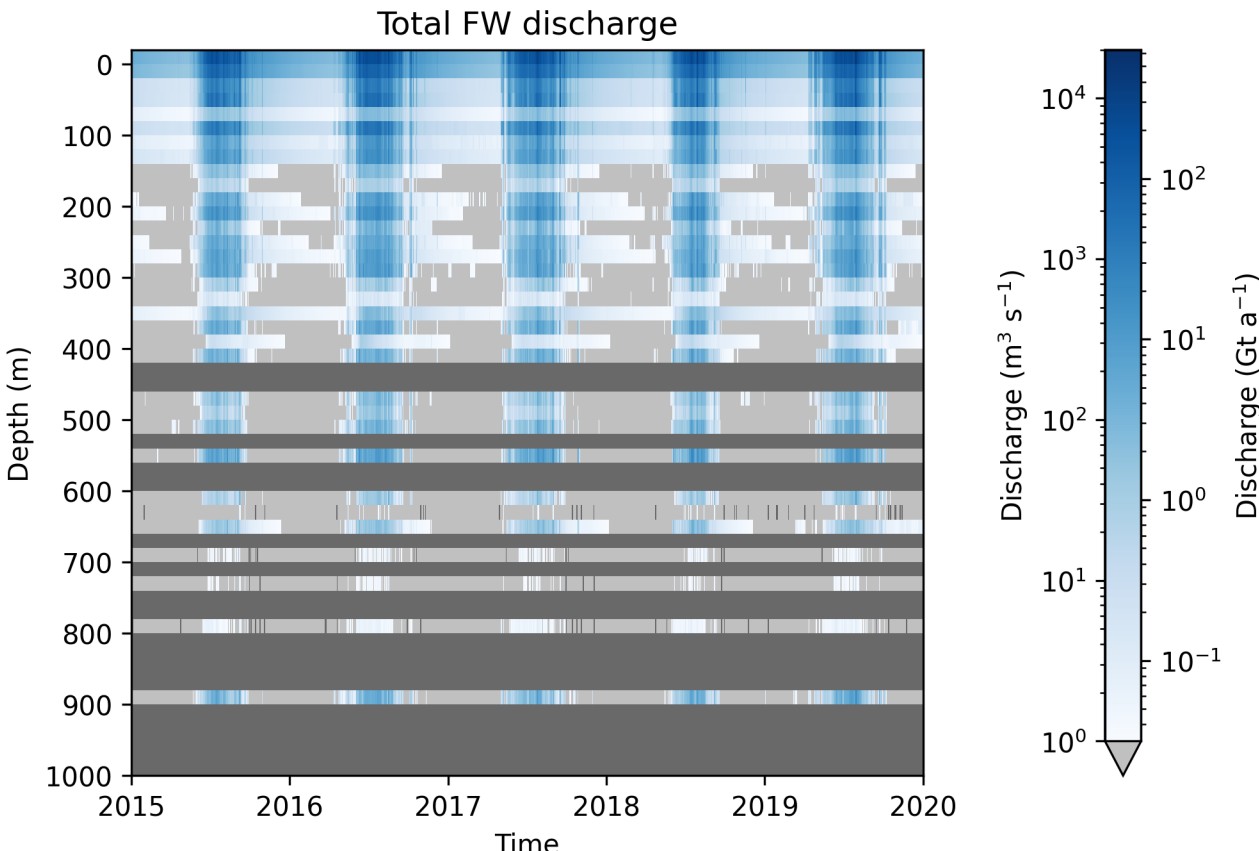


**Figure A1. Total freshwater (FW) discharge within SEG fjords during 2015 through 2019, representing only data within Mankoff**
**(2020) and Mankoff et al. (2020a). Freshwater discharge is binned into 20-m segments, from +20 − 0 m asl (above sea level) to 980 −**
**1000 m depth, with all discharge from elevations above 0 m asl included in the +20 − 0 m asl bin. Light gray areas indicate times**
**when the discharge in that bin was below a discharge threshold of 1 $m^3$ $s^{-1}$, while dark gray areas indicate no data were available.**



**Figure A2. Time series for fjord 18 (Kangerdlussuaq) showing: a) daily (black line) sea-ice area (km²) and percent coverage based on AMSR-2 sea ice concentration, along with a 31-day running mean (purple), b) area (km²) and percent coverage for fast ice evaluated from MODIS (blue dot) and Landsat (purple dot) single image sources and with smoothed (blue) record and for all four surface character types (0-3) for glacier-derived ice, c) total freshwater flux (m³ s⁻¹, black dashed line) and depth-binned (solid line) freshwater flux, d) cumulative fjord solid ice discharge (Gt yr⁻¹), and e) sea surface temperature (black line) and sea ice coverage (purple line) measured at the fjord mouth from MAR climate data. Vertical dashed orange lines in all panels indicate the freeze-up and break-up dates for offshore sea ice (panel a) as measured by passing a threshold of 15% of mean March-April sea ice area. A similar threshold is indicated (dashed line) in panel e, while panel b is a simple 15% threshold (dashed line). The 15% threshold is indicated by a dashed line in panels a, b, and e.**



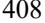

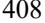

**Figure A3. Same as Fig. A2 for fjord 31 (Ikertivaq).**







**Figure A4. Same as Fig. A2 for fjord 37 (Skjoldungen), but with no solid ice discharge data and panel (e) presented as panel (d).**





**Figure A5. Same as Fig. A2 for fjord 40 (Timmiarmiut).**



**Figure A6. Same as Fig. A2 for fjord 43 (Naparsuaq).**





418
419 **Figure A7. Same as Fig. A2 for fjord 45 (Anoritoq).**



Figure A8. Same as Fig. A2 for fjord 48 (Kangerluluk), but with no solid ice discharge data and panel (e) presented as panel (d).



**Table A1. Statistics for landfast ice in SEG focus fjords. Using a threshold of 15% areal coverage to define the landfast ice season, each table entry contains the start day (day-of-year, doy), end day (doy), and duration (days) of the landfast ice season. Landfast ice analysis did not span the full 12-month year and < symbol indicates likely earlier presence while the > symbol indicates likely later/longer presence. Years when the landfast ice coverage never exceeded the 15% threshold are marked as ---. The last two columns give the mean and standard deviation of the start day (doy), end day (doy), and duration (days). Standard deviation is not calculated for records of likely longer length (> or < included). Dates are based on use of smoothed data (see section 3.3).**

| | | 2015 | 2016 | 2017 | 2018 | 2019 | Mean | Stdv |
|---|---|---|---|---|---|---|---|---|
| **Nansen** | Start day (doy) | <42 | <34 | <47 | <65 | <30 | <43.6 | |
| | End day (doy) | >179 | >159 | 175 | 148 | 144 | >161.1 | |
| | Duration (days) | >137 | >125 | >128 | >83 | >114 | >117.5 | |
| **Kangerdlugssuaq** | | <32 | <35 | <47 | <34 | <31 | <35.8 | |
| | | >182 | 157 | 158 | 158 | 142 | >159.3 | |
| | | >150 | >122 | >111 | >124 | >111 | >123.4 | |
| **Ikertivaq** | | 65 | <31 | 34 | 50 | <25 | <40.8 | |
| | | 160 | 124 | 137 | 119 | 116 | 131.1 | 18.0 |
| | | 95 | >93 | 103 | 69 | >91 | >90.2 | |
| **Skjoldungen** | | 52 | 28 | 62 | 25 | 31 | 39.6 | 16.3 |
| | | 148 | 163 | 124 | 148 | 120 | 140.4 | 18.3 |
| | | 96 | 135 | 62 | 123 | 89 | 100.8 | 28.8 |
| **Timmiarmiut** | | 30 | 11 | 52 | 35 | 43 | 34.0 | 15.5 |
| | | 134 | 164 | 120 | 145 | 159 | 144.3 | 18.0 |
| | | 104 | 153 | 68 | 110 | 116 | 110.3 | 30.4 |
| **Naparsuaq** | | 43 | 27 | 22 | 70 | --- | 40.3 | 21.6 |
| | | 147 | 156 | 52 | 151 | --- | 126.4 | 50.1 |
| | | 104 | 129 | 30 | 81 | --- | 86.1 | 42.2 |
| **Anoritoq** | | --- | 36 | 21 | 94 | 42 | 48.4 | 31.8 |
| | | --- | 148 | 130 | 127 | 117 | 130.4 | 13.3 |
| | | --- | 112 | 109 | 33 | 75 | 81.9 | 37.2 |
| **Kangerluluk** | | --- | 34 | 43 | 89 | --- | 55.3 | 29.5 |
| | | --- | 143 | 76 | 144 | --- | 120.8 | 39.0 |
| | | --- | 109 | 33 | 55 | --- | 65.6 | 38.7 |

**Code and data availability**

Data created to support this research is archived at the National Snow and Ice Data Center (Cohen et al., 2023).

The code for freshwater and solid ice discharge data analysis and visualization is available at

https://github.com/tarynblack/southeast_greenland_fjords [*This will be formally archived as a repository with DOI before final publication*].



Solid ice discharge data: v79 published 2023-05-05 at https://doi.org/10.22008/promice/data/ice_discharge/d/v02
Freshwater discharge data: v4.2 published 2022-08-28 at https://doi.org/10.22008/FK2/XKQVL7

**Author contributions and competing interests**

We used the CRediT taxonomy (https://casrai.org/credit/) to evaluate individuals' contributions and order authorship. All
authors designed the study and contributed to the writing and editing of the manuscript. TM and KL administrated the project
with TM supervising this research component. BC, TB, and HS were responsible for data collection and formal analyses. TM,
BC, TB, and HS validated data and produced data visualizations. IJ advised regarding early research methods.
The authors declare that they have no conflicts of interest.

**Acknowledgements**

This research was supported via NASA Biological Diversity and Ecological Forecasting Programs and Cryospheric Sciences
(NNX11AO63G, NNX13AN28G, and 80NSSC18K1229). We acknowledge Xavier Fettweis for assistance with MAR
regional climate model data and Brice Noël for assistance with RACMO regional climate model data (included in some
archived code and data but not within published results).

**References [_TO BE FORMATTED AFTER JOURNAL ACCEPTANCE_]**

Aschwanden, A., Fahnestock, M. A., Truffer, M., Brinkerhoff, D. J., Hock, R., Khroulev, C., Mottram, R., and Khan, S. A.:
Contribution of the Greenland Ice Sheet to sea level over the next millennium, 5, eaav9396,
https://doi.org/10.1126/sciadv.aav9396, 2019.
Beitsch, A., L. Kaleschke, and S. Kern (2014). Investigating High-Resolution AMSR2 Sea Ice Concentrations during the
February 2013 Fracture Event in the Beaufort Sea. Remote Sensing, 6, 3841-3856, doi:10.3390/rs6053841.
http://www.mdpi.com/2072-4292/6/5/3841.
Bochow, N., Poltronieri, A., Robinson, A., Montoya, M., Rypdal, M., and Boers, N.: Overshooting the critical threshold for
the Greenland ice sheet, Nature, 622, 528–536, https://doi.org/10.1038/s41586-023-06503-9, 2023.
Bosson, J. B., Huss, M., Cauvy-Frauníe, S., Clément, J. C., Costes, G., Fischer, M., Poulenard, J., and Arthaud, F.: Future
emergence of new ecosystems caused by glacial retreat, Nature, 620, 562–569, https://doi.org/10.1038/s41586-023-06302-2,

467  2023.



Cohen, B., T. Black, T. Moon. 2023. Southeast Greenland Fjord Physical Characteristics for 2015-2019. Boulder, Colorado
USA. NASA National Snow and Ice Data Center Distributed Active Archive Center. https://doi.org/
10.5067/R86BW8LR6PZH. (*Under Review*)
Fettweis, X., Box, J. E., Agosta, C., Amory, C., Kittel, C., Lang, C., As, D. V., Machguth, H., and Gallée, H.:
Reconstructions of the 1900–2015 Greenland ice sheet surface mass balance using the regional climate MAR model, 11,
1015–1033, https://doi.org/10.5194/tc-11-1015-2017, 2017.
Fox-Kemper, B., H.T. Hewitt, C. Xiao, G. Aðalgeirsdóttir, S.S. Drijfhout, T.L. Edwards, N.R. Golledge, M. Hemer,
R.E. Kopp, G. Krinner, A. Mix, D. Notz, S. Nowicki, I.S. Nurhati, L. Ruiz, J.-B. Sallée, A.B.A. Slangen, and Y. Yu, 2021:
Ocean, Cryosphere and Sea Level Change. In *Climate Change 2021: The Physical Science Basis. Contribution of Working*
*Group I to the Sixth Assessment Report of the Intergovernmental Panel on Climate Change* [Masson-Delmotte, V., P. Zhai,
A. Pirani, S.L. Connors, C. Péan, S. Berger, N. Caud, Y. Chen, L. Goldfarb, M.I. Gomis, M. Huang, K. Leitzell, E. Lonnoy,
J.B.R. Matthews, T.K. Maycock, T. Waterfield, O. Yelekçi, R. Yu, and B. Zhou (eds.)]. Cambridge University Press,
Cambridge, United Kingdom and New York, NY, USA, pp. 1211–1362, doi:10.1017/9781009157896.011.
Gallagher, M. R., Shupe, M. D., Chepfer, H., and L'Ecuyer, T.: Relating snowfall observations to Greenland ice sheet mass
changes: an atmospheric circulation perspective, Cryosphere, 16, 435–450, https://doi.org/10.5194/tc-16-435-2022, 2021.
Gelderloos, R., T.W.N. Haine, and M. Almansi (2022). Subinertial variability in four Southeast Greenland fjords in realistic
numerical simulations. Journal of Geophysical Research: Oceans, 127, e2022JC018820.
https://doi.org/10.1029/2022JC018820.
Hawkings, J. R., Linhoff, B. S., Wadham, J. L., Stibal, M., Lamborg, C. H., Carling, G. T., Lamarche-Gagnon, G., Kohler,
T. J., Ward, R., Hendry, K. R., Falteisek, L., Kellerman, A. M., Cameron, K. A., Hatton, J. E., Tingey, S., Holt, A. D.,
Vinšová, P., Hofer, S., Bulínová, M., Větrovský, T., Meire, L., and Spencer, R. G. M.: Large subglacial source of mercury
from the southwestern margin of the Greenland Ice Sheet, Nat Geosci, 14, 496–502, https://doi.org/10.1038/s41561-021-
00753-w, 2021.
Heide-Jørgensen, M. P., Chambault, P., Jansen, T., Gjelstrup, C. V. B., Rosing-Asvid, A., Macrander, A., Víkingsson, G.,
Zhang, X., Andresen, C. S., and MacKenzie, B. R.: A regime shift in the Southeast Greenland marine ecosystem, Global
Change Biol, https://doi.org/10.1111/gcb.16494, 2022.
Hersbach, H. *et al.* The ERA5 global reanalysis. *Q. J. R. Meteorol. Soc.* **146**, 1999–2049 (2020).
Hopwood, M. J., Carroll, D., Browning, T. J., Meire, L., Mortensen, J., Krisch, S., and Achterberg, E. P.: Non-linear
response of summertime marine productivity to increased meltwater discharge around Greenland, 1–9,
https://doi.org/10.1038/s41467-018-05488-8, 2018.



Hopwood, M. J., Carroll, D., Dunse, T., Hodson, A., Holding, J. M., Iriarte, J. L., Ribeiro, S., Achterberg, E. P., Cantoni, C.,
Carlson, D. F., Chierici, M., Clarke, J. S., Cozzi, S., Fransson, A., Juul-Pedersen, T., Winding, M. H. S., and Meire, L.:
Review article: How does glacier discharge affect marine biogeochemistry and primary production in the Arctic?, 14, 1347–
1383, https://doi.org/10.5194/tc-14-1347-2020, 2020.
Kaleschke, L. and X. Tian-Kunze (2016). AMSR2 ASI 3.125 km Sea Ice Concentration Data, V0.1, Institute of Oceanography,
University of Hamburg, Germany, digital media (ftp-projects.zmaw.de/seaice/).
Kelly, B. P., J. L. Bengtson, P. L. Boveng, M. F. Cameron, S. P. Dahle, J. K. Jansen, E. A. Logerwell, J. E. Overland, C. L.
Sabine, G. T. Waring, and J. M. Wilder 2010. Status review of the ringed seal (Phoca hispida). U.S. Dep. Commer., NOAA
Tech. Memo. NMFS-AFSC-212, 250 p.
Kim, Y.-H., Min, S.-K., Gillett, N. P., Notz, D., and Malinina, E.: Observationally-constrained projections of an ice-free
Arctic even under a low emission scenario, Nat. Commun., 14, 3139, https://doi.org/10.1038/s41467-023-38511-8, 2023.
Kochtitzky, W. & Copland, L. Retreat of Northern Hemisphere Marine-Terminating Glaciers, 2000–2020. *Geophys Res Lett*
**49**, (2022).
Laidre, K. L. and I. Stirling. 2020. Grounded icebergs as maternity denning habitat for polar bears (Ursus maritimus) in
North and Northeast Greenland. Polar Biology 43(7): 937-943, 10.1007/s00300-020-02695-2.
Laidre, K. L., Supple, M. A., Born, E. W., Regehr, E. V., Wiig, Ø., Ugarte, F., Aars, J., Dietz, R., Sonne, C., Hegelund, P.,
Isaksen, C., Akse, G. B., Cohen, B., Stern, H. L., Moon, T., Vollmers, C., Corbett-Detig, R., Paetkau, D., and Shapiro, B.:
Glacial ice supports a distinct and undocumented polar bear subpopulation persisting in late 21st-century sea-ice conditions,
Science, **376**, 1333–1338, https://doi.org/10.1126/science.abk2793, 2022.
Lenaerts, J. T. M., Medley, B., Broeke, M. R., and Wouters, B.: Observing and Modeling Ice Sheet Surface Mass Balance,
Reviews Of Geophysics, 57, 376–420, https://doi.org/10.1029/2018rg000622, 2019.
Mahoney, A. R., Eicken, H., Gaylord, A. G., and Gens, R.: Landfast sea ice extent in the Chukchi and Beaufort Seas: The
annual cycle and decadal variability, Cold Reg Sci Technol, 103, 41–56, https://doi.org/10.1016/j.coldregions.2014.03.003,
521    2014.

Mankoff, K.: "Greenland freshwater runoff, GEUS Dataverse, V2, https://doi.org/10.22008/FK2/AA6MTB, 2020.
Mankoff, K. D., Noël, B., Fettweis, X., Ahlstrøm, A. P., Colgan, W., Kondo, K., Langley, K., Sugiyama, S., van As, D., and
Fausto, R. S.: Greenland liquid water discharge from 1958 through 2019, Earth Syst. Sci. Data, 12, 2811–2841,
https://doi.org/10.5194/essd-12-2811-2020, 2020a.
Mankoff, K. D.; Solgaard, A., Larsen, S.: Greenland Ice Sheet solid ice discharge from 1986 through last month: Discharge,
GEUS Dataverse, V54, https://doi.org/10.22008/promice/data/ice_discharge/d/v02, 2020b.



Mankoff, K. D., Solgaard, A., Colgan, W., Ahlstrøm, A. P., Khan, S. A., and Fausto, R. S.: Greenland Ice Sheet solid ice
discharge from 1986 through March 2020, Earth Syst. Sci. Data, 12, 1367–1383, https://doi.org/10.5194/essd-12-1367-2020,
2020c.
McGovern, M., A. E. Poste, E. Oug, P. E. Renaud, H. C. Trannum. Riverine impacts on benthic biodiversity and functional
traits: A comparison of two sub-Arctic fjords. *Estuarine, Coastal and Shelf Science* **240**, (2020),
https://doi.org/10.1016/j.ecss.2020.106774.
Meire, L., Paulsen, M. L., Meire, P., Rysgaard, S., Hopwood, M. J., Sejr, M. K., Stuart-Lee, A., Sabbe, K., Stock, W., and
Mortensen, J.: Glacier retreat alters downstream fjord ecosystem structure and function in Greenland, Nat. Geosci., 16, 671–
674, https://doi.org/10.1038/s41561-023-01218-y, 2023.
Moon, T. A., Gardner, A. S., Csatho, B., Parmuzin, I., and Fahnestock, M. A.: Rapid reconfiguration of the Greenland Ice
Sheet coastal margin, 1–25, https://doi.org/10.1029/2020jf005585, 2020.
Moon, T. A., K. D. Mankoff, R. S. Fausto, X. Fettweis, B. D. Loomis, T. L. Mote, K. Poinar, M. Tedesco, A. Wehrlé, and C.
D. Jensen, 2022: Greenland Ice Sheet. *Arctic Report Card 2022*, M. L. Druckenmiller, R. L. Thoman, and T. A. Moon, Eds.,
https://doi.org/10.25923/c430-hb50.
Morlighem, M., C. Williams, E. Rignot, L. An, J. E. Arndt, J. Bamber, G. Catania, N. Chauché, J. A. Dowdeswell, B.
Dorschel, I. Fenty, K. Hogan, I. Howat, A. Hubbard, M. Jakobsson, T. M. Jordan, K. K. Kjeldsen, R. Millan, L. Mayer, J.
Mouginot, B. Noël, C. O'Cofaigh, S. J. Palmer, S. Rysgaard, H. Seroussi, M. J. Siegert, P. Slabon, F. Straneo, M. R. van den
Broeke, W. Weinrebe, M. Wood, and K. Zinglersen. 2017. BedMachine v3: Complete bed topography and ocean bathymetry
mapping of Greenland from multi-beam echo sounding combined with mass conservation. *Geophysical Research Letters*.
44. DOI: 10.1002/2017GL074954.
Morlighem, M. et al. (2022). IceBridge BedMachine Greenland, Version 5 [Data Set]. Boulder, Colorado USA. NASA
National Snow and Ice Data Center Distributed Active Archive Center. https://doi.org/10.5067/GMEVBWFLWA7X. Date
Accessed 11-02-2023.
Noël, B., Berg, W. J. van de, Science, S. L., 2019: Rapid ablation zone expansion amplifies north Greenland mass loss,

552    2019.

Petrich, C., Eicken, H., Zhang, J., Krieger, J., Fukamachi, Y., and Ohshima, K. I.: Coastal landfast sea ice decay and breakup
in northern Alaska: Key processes and seasonal prediction, Journal of Geophysical Research, 117, C02003,
https://doi.org/10.1029/2011jc007339, 2012.
Rastner, P., Bolch, T., Mölg, N., Machguth, H., Bris, R. L., and Paul, F.: The first complete inventory of the local glaciers
and ice caps on Greenland, The Cryosphere, 6, 1483–1495, https://doi.org/10.5194/tc-6-1483-2012, 2012.



Scheick, J., Enderlin, E. M., and Hamilton, G.: Semi-automated open water iceberg detection from Landsat applied to Disko
Bay, West Greenland, Journal Of Glaciology, 65, 468–480, https://doi.org/10.1017/jog.2019.23, 2019.
Soldal, I., Dierking, W., Korosov, A., and Marino, A.: Automatic Detection of Small Icebergs in Fast Ice Using Satellite
Wide-Swath SAR Images, Remote Sensing, 11, 806–24, https://doi.org/10.3390/rs11070806, 2019.
Stern, H. L. and Laidre, K. L.: Sea-ice indicators of polar bear habitat, The Cryosphere, 10, 2027–2041,
https://doi.org/10.5194/tc-10-2027-2016, 2016.
Dirk van As, Bent Hasholt, Andreas P. Ahlstrøm, Jason E. Box, John Cappelen, William Colgan, Robert S. Fausto, Sebastian
H. Mernild, Andreas Bech Mikkelsen, Brice P.Y. Noël, Dorthe Petersen & Michiel R. van den Broeke (2018) Reconstructing
Greenland Ice Sheet meltwater discharge through the Watson River (1949–2017), Arctic, Antarctic, and Alpine
Research, 50:1, DOI: 10.1080/15230430.2018.1433799
van Dongen, E. C. H., Jouvet, G., Sugiyama, S., Podolskiy, E. A., Funk, M., Benn, D. I., Lindner, F., Bauder, A., Seguinot,
J., Leinss, S., and Walter, F.: Thinning leads to calving-style changes at Bowdoin Glacier, Greenland, The Cryosphere, 15,
485–500, https://doi.org/10.5194/tc-15-485-2021, 2021.
White, D.R. Propagation of Uncertainty and Comparison of Interpolation Schemes. *Int J Thermophys* **38**, 39 (2017).
https://doi.org/10.1007/s10765-016-2174-6