# Peer review of "freshwater flux to support biological applications"

_EGUsphere, 2024_

## Referee Comment (RC2)

The author presents a comprehensive study on the Southeast Greenland (SEG) fjord systems, focusing on surface ice conditions and freshwater flux, especially differentiating the glacier-derived and Landfast ice from Landsat-8 and MODIS. The manuscript is well-structured, providing a significant reference for understanding the possible dynamic habitats in the context of ongoing climate change. But, I do have some concerns about the satellite algorithm, uncertainty quantification, and interpretation of the data in the following:

(1) When I saw the title, I thought the authors would include sections connecting the surface ice and freshwater flux with the biological system. But I didn't find that part, only some future implications, which made the title a little disconnected without hitting the main point of the paper.

(2) Line 88-93. I don't know why you chose Skioldungen and Kangerluluk since, from Figure 1, those don't have the glacier location upstream, which also compromises the robustness of the characteristics of the SEG fjord. (I knew they are heavily occupied by polar bears, but since you want to link the physical and biological fjord system, doesn't it make sense to consider the glacier and polar bear activities as well?)

(3) Line 157: Can I have more details on why you chose 15% as your threshold? Does it make a large difference in differentiating the transition period in Figure 6?

(4) Line 163-170: After reading the paper, I hardly found an explanation for why you also analyze the SST and sea ice coverage in Figure 9 (e) since you didn't mention their changes and connection with the Fjord system. If you want to show the MAR performance in those parameters, I would suggest you put them in the Appendix.

(5) Section 3.4. I have the same concerns as Reviewer 1 since Landsat-8 and MODIS are primarily limited and impeded by clouds and water vapour. How many of the images are really affected by the cloud? The visual classification doesn't consider the cloud, right? Then, how much real MODIS and Landsat-8 data are used in the visual classification compared to the full-sky MODIS and Landsat-8 before cloud filtering? Can we take the filtered data as well-representation? Does it make the results have a systematic bias?

(6) Line 205-209. I know visual classification is quite time-consuming work, and you've analyzed a lot of MODIS and Landsat-8 data, but can I know the satellite passing by date between two satellites? Further questions on how to clarify the potential limitations or biases introduced by manual digitization and how they were mitigated would add to the methodological rigour. How can the uncertainties from MODIS and Landsat-8 be quantified since Table 2 seems quite dependent on individual experiences?

(7) Figure 10. What's the spatial resolution of frequencies in fast ice presence and glacial ice presence?

(8) I am slightly lost when you show solid ice discharge, landfast ice, and Glacier ice values. Can you explain their possible implications? Since, for me, they are just shown here without any inner interpretation or further analysis. The integration of diverse datasets is a strength of this study. However, discussing the challenges and uncertainties involved in combining different data types (e.g., remote

sensing data with climate model outputs) and how these might affect the interpretation of results would be beneficial. Moreover, a more detailed analysis of the variability observed across the fjords and its potential biological implications would add depth to the study.

(9) As I mentioned in the first point. The manuscript misses a deeper discussion on the specific biological applications of the physical environment characterization provided. For instance, detailing potential impacts on the habitat preferences, migration patterns, or population dynamics of key species would directly link the physical and biological aspects of the fjord systems.

---

## Author Comment (AC1)

Thank you for helpful comments that have improved the manuscript. The full content of comments and responses to Anonymous Reviewer #1 are below.

Reviewer Comments in **black** and responses in **red**

Anonymous Reviewer #1

The paper "Characterizing Southeast Greenland fjord surface ice and freshwater flux to support biological applications" is an interesting and topical read. It's primarily a physical and data analysis paper concerning ice dynamics in the inshore regions of SE Greenland but is written for, and potentially to help, a more biologically orientated research audience. I think this an interesting concept and worthwhile effort. As a more marine/biology orientated reader I hope the following comments are useful. I will defer to other reviewers concerning any technical aspects of data processing because I cannot comment on these in much detail- especially the use of satellite imagery. I think the manuscript is suitable for The Cryosphere and will be a much more interesting read and resource for a broader audience than the raw data products themselves.

Thank you. We also anticipate the paper will be of interest to a multidisciplinary audience.

12 with a focus

Changed

34 Is there a specific reason why Polar Bears are flagged as a species of interest? Perhaps because of the recent highlighting of a population permanently resident in SE Greenland? If I was going to pick one mammal of regional relevance I would probably have picked the narwhal as this region hosts a protected zone specifically for a narwhal population.

Polar bears are flagged as the species of interest because of their significant use of the fjord surface ice and the importance of this physical environment to polar bear life functions. This area is home to a genetically distinct resident subpopulation. A subset of the data in this paper has already been used for comparison with polar bear behavior (Laidre et al., 2022). Narwhals are not a primary species of interest in this region, contrary to the reviewer's suggestion. Narwhals occur further north (around Tasiilaq and the Blosseville coast), but not regularly south of 64N. However, the data are available to combine with any other biological data of a user's choice.

46 'primary productivity' It is important to distinguish between primary and secondary productivity as this seems to be the source of a lot of confusion in the literature in comments about 'highly productive' fjords, e.g. glacier fjords with low primary production can sometimes act as regionally significant hotspots for bird and mammal populations for

reasons other than high primary production driving a disconnection between the two which is often muddled.

We have changed to "primary productivity".

47 I would not suggest raising Hg as a topical issue, the Hg data from Hawkings et al., 2021 is dubious, other groups report Hg numbers 1000 times lower than claimed in the same location (Jørgensen et al., 2024) and a recent preprint in SE Greenland shows the same low levels (https://www.researchsquare.com/article/rs-3289576/v1).  The consensus opinion of the Arctic Hg community is that the Hawkings et al., 2021 values are likely erroneous and, without further verification, the Greenland Ice Sheet does not constitute a major Hg source (Dastoor et al., 2022). I would suggest high turbidity is a much less controversial and more commonly recognized issue of biological relevance to highlight e.g. (Holding et al., 2019; Murray et al., 2015; Sejr et al., 2022)

Thank you for this additional information. We have changed the references to remove the Hawkings et al. 2021 paper and instead include the three suggested references, which we agree do provide useful information on the influence of meltwater in Greenland fjords.

49 Suggest 'rapid physical changes' unless providing references also showing time-series effects on biology (I don't think there are many)

Changed as suggested.

55 Ice presence will also affect light availability, a major driver of primary production, perhaps worth mentioning here

Good point. We have added "Surface ice presence may also alter other factors, such as light availability in the water column, salinity, or ocean water mixing, that may be of interest to other biological researchers."

3.1 It's very hard to visualize all this information and assess what is/is not important to data quality. Obviously, there are general issues which the authors' have worked hard to address and conclude do not impact the conclusions, could a figure, perhaps a supplementary one, show the data availability for each system and better convey this information (e.g. as per Figure 3)?

As suggested, we have added a new figure to the appendix that shows the solid ice discharge observation availability, with a style aligning with Figure 3.

3.2 Could the authors define here (or earlier) what freshwater is and what it does, and does not include? Is it basically the Mankoff definition? And if so, somewhere for clarity could the

authors' clarify what freshwater would be/not be included within these estimates/fluxes e.g. is runoff not originating from the Ice Sheet included and glaciers not connected to the ice sheet? I assume given the topography of the region these are assumed to be a minor freshwater source, is this the case for all of the case studies?

We do use the Mankoff et al. (2020; 2020a) definition for freshwater flux and have added that information to this section. We also clarify that peripheral glaciers may be excluded from this input dataset (because they are not included in the regional climate model ice domain), though these features are scarce within our region of interest.

150- Not an expert on this so forgive a query if it's not the case – is this still affected by cloud cover to some extent and if so how problematic is the interference, is it quantified/quantifiable?

The freshwater flux data are derived from regional atmospheric climate model data, so optical satellite imagery is not an input to this dataset and has no influence on freshwater flux time series.

175- Same query, can the authors give some rough numbers for the loss of data due to cloud cover and over-winter.

Figure 3 gives a visual reference regarding the amount of usable imagery that was available for each fjord across the time period of interest and we believe this is an ideal reference for within the manuscript. Below is also a table showing the number of analyzed images for each fjord and year during January 1 – July 1 (landfast ice; Landsat and MODIS) or the full year (glacial ice; Landsat only). While this time periods might suggest that there are ~182 image days for possible landfast ice analysis, optical imagery is limited not only by cloud cover, but also by polar night, so both influence the number of usable images noted here (also true for glacial ice, but Landsat 8 does not have daily coverage). We also did not analyze images for landfast ice after we observed the arrival of ice-free conditions.

**Landfast Ice**

| Fjord | 2015 | 2016 | 2017 | 2018 | 2019 |
|---|---|---|---|---|---|
| 15 | 51 | 53 | 59 | 38 | 51 |
| 18 | 55 | 52 | 44 | 43 | 46 |
| 31 | 30 | 36 | 31 | 27 | 32 |
| 37 | 59 | 48 | 42 | 48 | 34 |
| 40 | 62 | 47 | 51 | 47 | 45 |
| 43 | 61 | 45 | 58 | 59 | 32 |
| 45 | 65 | 46 | 58 | 71 | 30 |
| 48 | 59 | 43 | 47 | 63 | 33 |

**Glacial Ice**

| Fjord | 2015 | 2016 | 2017 | 2018 | 2019 |
|---|---|---|---|---|---|
| 15 | 17 | 18 | 16 | 21 | 20 |
| 18 | 15 | 14 | 13 | 17 | 16 |
| 31 | 8 | 9 | 8 | 3 | 5 |
| 37 | 9 | 8 | 12 | 10 | 6 |
| 40 | 10 | 8 | 12 | 8 | 6 |
| 43 | 15 | 10 | 15 | 16 | 9 |
| 45 | 17 | 17 | 17 | 19 | 13 |
| 48 | 15 | 11 | 9 | 12 | 12 |

193 To clarify, this was done visually right? The analyst is manually tagging ice cover as bergy bits/glacial ice, landfast and pack ice floes?

This is correct and we have modified to "visual digitization process".

Figure 3 concerns the raw data or the processed data? e.g. if a MODIS image was available, but had 100% cloud cover, would it be plotted on Fig. 3? Maybe the figure could be improved a little if the dots reflected the quality/usefulness of the data as well e.g. shading out MODIS data with heavy cloud cover?

The figure shows the final data used in analysis, including only quality data. We have modified the caption to make this clear. We do not further distinguish on image quality, as all images included in the figure are used in analysis.

208 The % change might be more useful to quote as it's hard to understand how large an error this is?

We appreciate the suggestion but have not changed the text. The reason for this is that differences in digitization are all about visual edges and edge resolutions. In other words,

the agreement between a certain area digitized using MODIS vs Landsat will differ not by a percentage of the total fjord area but rather a "fixed" area in regard to the boundaries one could trace in imagery from each source – it is independent of fjord area. We note here that the areas stated represent areas of less than 1% of the individual fjord areas.

337 Just for clarity, I think the authors' implication here is that these areas' bathymetry is poorly mapped because the areas are likely very shallow? Maybe edit accordingly.

The following sentence states: "Comparing landfast ice locations with bathymetric data from BedMachine 5 (Morlighem et al., 2017; Morlighem et al., 2022), for example, landfast ice often occurs in presumably shallow regions that lack any bathymetric detail." So, we feel that the reviewer's request is already addressed within the text.

337-345 Is there potentially a water mass effect here as well? Heat for ice melt comes mainly from the inflow of warm Atlantic water at depth (Straneo & Cenedese, 2015), so shallow areas occupied by a flow of Polar Water and cut off from the main estuarine circulation of a fjord might experience different heat budgets?

We agree that bathymetry influences water mass access within fjords and local heat budgets. The depths we discuss in this section are likely, however, to be on the order of 0-50 meters, much shallower than the expected depth for warm Atlantic inflows, so we expect minimal changes in deeper warm water access for these unmapped areas. We have left the text as is, highlighting surface wave processes and physical bathymetry high points that could ground landfast ice.

351 Reference for seals? My limited understanding was that the evidence for seal-ice associations in glacier fjords is mixed and maybe species specific and regionally dependent (Womble et al., 2021)

The current Laidre et al. 2022 reference does report on predation events with polar bears eating seals in SE Greenland. Due to the difficulty of research in SE Greenland, there are no direct peer-reviewed papers for seals in the region. The Womble et al. 2021 paper focused on Alaska (as does Kelly et al. 2010, which we cite elsewhere in the paper). Another general reference (Lydersen et al. 2014) focuses on Svalbard. There is a grey literature report (Boertmann and Rosing-Asvid, 2014) from a SE Greenland bird and seal survey, but they were not able to survey in ice-covered regions. This led them to conclude: "The relatively high fraction of bearded seals in the water indicates that our route (in open water areas along the coast) did not include the main areas for bearded seals at that time of the year. Bearded seals seek out ice for haul-out in July and they were also seen in the patches of ice

with many hooded seals. Surveys into the densely packed ice-fjords would probably reveal higher concentrations of bearded seals in July."

We have left the text unchanged.

Christian Lydersen, Philipp Assmy, Stig Falk-Petersen, Jack Kohler, Kit M. Kovacs, Marit Reigstad, Harald Steen, Hallvard Strøm, Arild Sundfjord, Øystein Varpe, Waldek Walczowski, Jan Marcin Weslawski, Marek Zajaczkowski (2014), The importance of tidewater glaciers for marine mammals and seabirds in Svalbard, Norway, Journal of Marine Systems, Volume 129, Pages 452-471, ISSN 0924-7963, https://doi.org/10.1016/j.jmarsys.2013.09.006.

David Boertmann, Aqqalu Rosing-Asvid (2014), Seabirds and seals in southeast Greenland: Results from a survey in July 2014, Scientific report from DCE – Danish Centre for Environment and Energy, No. 117, https://dce2.au.dk/pub/SR117.pdf.

358 I assume, maybe the authors can clarify, that wind-based products are simply not available for fjord regions because there's no in situ monitoring and coastal productions cannot be meaningfully extrapolated?

Regional climate models (such as MAR, which we used) can provide gridded wind variables (direction, speed). We have previously done some comparisons between wind products and the few in situ weather stations in Southeast Greenland. While providing reasonable agreements, we ultimately decided not to include wind data within this study. Others may be interested to pursue this idea further.

359-366 Might some overview comments about areas be useful for the reader, what sort of total area and fractional areas of the fjords have each ice type?

The full time series for these metrics are provided within the multi-panel time series that are included for each fjord (one figure within the main text and the other seven figures within the Appendix). Together, these show both area of coverage and percent fjord cover for landfast ice and for all categories of glacier-derived ice. There's high interannual variability, so we prefer that readers refer to these for complete data rather than including a summary in the text.

379 'well underway', a specific statistic and reference to the historical record might be better

We agree that better wording is needed. We now say "With ongoing sea-ice loss along the east coast of Greenland (Stern and Laidre, 2016)...". The reviewer might be interested to know that the sea-ice loss numbers from this paper have actually been updated in 2023

and reported within the Polar Bear Specialist Group Status Table (https://www.iucn-pbsg.org/population-status/) for East Greenland (-5.7 days per decade for change in spring sea-ice retreat and +7.0 days per decade for change in date of fall sea-ice advance), but we don't think this source is an allowable citation for the paper.

Figure A1 I don't understand the dark bars. These values are basically using Mankoff discharge data with glacier grounding line depth from bedmachine right? So I read the dark grey area from 900-1000 m for all years to mean 'no data were available', which would imply to me that there were grounding lines in this depth range with no discharge data. Is this correct, or is it rather the case that there are no grounding lines in this depth range so the value is 0?

We reviewed the various use of zero, near-zero, and NaN values within the Mankoff (2020) and Mankoff et al. (2020a) datasets and stemming from the MAR and RACMO regional climate models. Based on this, we have updated the figure to ensure that light grey regions reflect the presence of discharge outlets but with discharge below our minimum threshold, while dark grey regions indicate that no outlets are present at those depths. The figure and caption are updated accordingly.

References referred to:

Dastoor, A., Angot, H., Bieser, J., Christensen, J. H., Douglas, T. A., Heimbürger-Boavida, L.-E., Jiskra, M., Mason, R. P., McLagan, D. S., Obrist, D., Outridge, P. M., Petrova, M. V, Ryjkov, A., St. Pierre, K. A., Schartup, A. T., Soerensen, A. L., Toyota, K., Travnikov, O., Wilson, S. J., & Zdanowicz, C. (2022). Arctic mercury cycling. *Nature Reviews Earth & Environment*. https://doi.org/10.1038/s43017-022-00269-w

Holding, J. M., Markager, S., Juul-Pedersen, T., Paulsen, M. L., Møller, E. F., Meire, L., & Sejr, M. K. (2019). Seasonal and spatial patterns of primary production in a high-latitude fjord affected by Greenland Ice Sheet run-off. *Biogeosciences*. https://doi.org/10.5194/bg-16-3777-2019

Jørgensen, C. J., Søndergaard, J., Larsen, M. M., Kjeldsen, K. K., Rosa, D., Sapper, S. E., Heimbürger-Boavida, L.-E., Kohler, S. G., Wang, F., Gao, Z., Armstrong, D., & Albers, C. N. (2024). Large mercury release from the Greenland Ice Sheet invalidated. *Science Advances*, *10*(4), eadi7760. https://doi.org/10.1126/sciadv.adi7760

Murray, C., Markager, S., Stedmon, C. A., Juul-Pedersen, T., Sejr, M. K., & Bruhn, A. (2015). The influence of glacial melt water on bio-optical properties in two contrasting Greenlandic fjords. *Estuarine, Coastal and Shelf Science*, *163*(PB), 72–83. https://doi.org/10.1016/j.ecss.2015.05.041

Sejr, M. K., Bruhn, A., Dalsgaard, T., Juul-Pedersen, T., Stedmon, C. A., Blicher, M., Meire, L., Mankoff, K. D., & Thyrring, J. (2022). Glacial meltwater determines the balance between autotrophic and heterotrophic processes in a Greenland fjord. *Proceedings of the National Academy of Sciences*, *119*(52), e2207024119. https://doi.org/10.1073/pnas.2207024119

Straneo, F., & Cenedese, C. (2015). The Dynamics of Greenland's Glacial Fjords and Their Role in Climate. *Annual Review of Marine Science*, *7*, 89–112. https://doi.org/10.1146/annurev-marine-010213-135133

Womble, J. N., Williams, P. J., McNabb, R. W., Prakash, A., Gens, R., Sedinger, B. S., & Acevedo, C. R. (2021). Harbor Seals as Sentinels of Ice Dynamics in Tidewater Glacier Fjords. *Frontiers in Marine Science*, *8*. https://www.frontiersin.org/articles/10.3389/fmars.2021.634541

---

## Author Comment (AC2)

Thank you for helpful comments that have improved the manuscript. The full content of comments and responses for Anonymous Reviewer #2 are below.

Reviewer Comments in **black** and responses in **red**

Anonymous Reviewer #2

The author presents a comprehensive study on the Southeast Greenland (SEG) fjord systems, focusing on surface ice conditions and freshwater flux, especially differentiating the glacier-derived and Landfast ice from Landsat-8 and MODIS. The manuscript is well-structured, providing a significant reference for understanding the possible dynamic habitats in the context of ongoing climate change. But, I do have some concerns about the satellite algorithm, uncertainty quantification, and interpretation of the data in the following:

(1) When I saw the title, I thought the authors would include sections connecting the surface ice and freshwater flux with the biological system. But I didn't find that part, only some future implications, which made the title a little disconnected without hitting the main point of the paper.

The manuscript intent is to provide a physical science basis for ongoing and future biological applications. The paper metrics and methods were created via collaboration between physical and biological scientists and we note previously published research that uses a subset of these data to gain further knowledge of Southeast Greenland polar bears. The intent of this paper is not make further connections between physical and biological conclusions within this publication, but rather to create and share a physical science fjord ice analysis that was designed for biological applications. We feel the paper title reflects that intent, but are open to suggestions for alternative titles.

(2) Line 88-93. I don't know why you chose Skioldungen and Kangerluluk since, from Figure 1, those don't have the glacier location upstream, which also compromises the robustness of the characteristics of the SEG fjord. (I knew they are heavily occupied by polar bears, but since you want to link the physical and biological fjord system, doesn't it make sense to consider the glacier and polar bear activities as well?)

These fjords are included precisely because they help to create a wide range of fjord environments with varying levels of glacial ice input and areas that include varying polar bear use.

(3) Line 157: Can I have more details on why you chose 15% as your threshold? Does it make a large difference in differentiating the transition period in Figure 6?

Figure 9a is a time series of the offshore sea-ice area at the mouth of fjord #15. Our objective is to detect the start of the sea-ice season in the fall and the end of the sea-ice season in the spring by choosing a sea-ice area threshold and finding the dates when the time series of sea-ice area crosses that threshold. We want to choose a relatively low threshold to closely detect the timing of sea-ice appearance (fall) and disappearance (spring), but not too low that noise or small fluctuations affect the timing. The threshold needs to be scaled or normalized to something, and we've chosen to scale it to the mean March-April sea-ice area, which is set at 100%. The choice of 15% of that value for the threshold is somewhat arbitrary, but it meets the conditions of being relatively low while still excluding noise and small fluctuations from detection. The resulting interval of time between the disappearance of sea-ice in the spring and the re-appearance of sea-ice in the fall, marked by the vertical dashed orange lines in all panels of Figure 9, matches reasonably well with the period of time when the sea surface temperature (SST) is above the freezing point of -1.8 degrees C (panel e). That gives us confidence that the 15% threshold is a reasonable choice. A reader can also view the full time series within our figures if they would like to consider different thresholds.

Regarding the effect of the threshold on the spring and fall transition dates shown in Figure 6, choosing a threshold larger than 15% would cause the spring dates to shift to the left (earlier) and the fall dates to shift to the right (later), but the shapes of the curves would not change much. Also, since the slope of the sea-ice area time series is relatively steep at the times when it crosses the threshold, a small change in the threshold would lead to a very small change in the crossing dates.

(4) Line 163-170: After reading the paper, I hardly found an explanation for why you also analyze the SST and sea ice coverage in Figure 9 (e) since you didn't mention their changes and connection with the Fjord system. If you want to show the MAR performance in those parameters, I would suggest you put them in the Appendix.

The reviewer is correct that didn't mention the MAR SST or sea-ice coverage in Figure 9e outside of the description at lines 163-170. We have added the following text to the revised manuscript in section 4.2: "Finally, we compared the spring and fall sea ice transition dates as calculated from AMSR2 sea-ice coverage (e.g. Fig 9a, vertical dashed orange lines) vs. MAR sea-ice coverage (e.g. Fig 9e, vertical dashed blue lines) for the eight focus fjords. For the three northern fjords (#15, 18, 31), which are all north of 64N, the agreement is quite good: the mean dates (across 5 years) are within 3 days of each other. These fjords have relatively well-defined annual cycles of sea-ice coverage, so there is little ambiguity in identifying the transition dates. For the five southern fjords, which are all south of 64N, the

agreement is less good: mean differences can be as high as +/-2 weeks, with larger variability than for the northern fjords. These fjords have relatively choppy annual cycles of sea-ice coverage, with lots of spikes, so the detection of the transition dates is noisier."

Regarding whether Figure 9e belongs in the Appendix, we note that the MAR data for all the other fjords are already in the Appendix (Figures A2-A8, bottom panel). We don't think it's necessary or desirable to break up Figure 9 and put panel (e) in the Appendix by itself. The sea surface temperature (SST) and offshore sea ice coverage data are included in the multi-panel time series because we are endeavoring to provide more of a *systems* characterization of the Southeast Greenland fjord environment, so it is important that these metrics are viewable together.

(5) Section 3.4. I have the same concerns as Reviewer 1 since Landsat-8 and MODIS are primarily limited and impeded by clouds and water vapour. How many of the images are really affected by the cloud? The visual classification doesn't consider the cloud, right? Then, how much real MODIS and Landsat-8 data are used in the visual classification compared to the full-sky MODIS and Landsat-8 before cloud filtering? Can we take the filtered data as well-representation? Does it make the results have a systematic bias?

As noted in response to Reviewer #1, we have clarified the information in the Figure 3 caption to make clear that only imagery determined to have clear conditions over the fjord analysis area are included in the figure and in later analysis. This determination is made directly within our team; we do not depend on product cloud classifications to do this since those can run into problems distinguishing across clouds, ice, etc. In this way, clouds are removed before fjord surface digitization is undertaken. This does limit the amount of data that was usable for analysis and Figure 3 clearly shows the final distribution of data used. There may be a systematic bias related to data reduction due to polar night, but we do not identify a systematic bias due to cloud cover.

(6) Line 205-209. I know visual classification is quite time-consuming work, and you've analyzed a lot of MODIS and Landsat-8 data, but can I know the satellite passing by date between two satellites? Further questions on how to clarify the potential limitations or biases introduced by manual digitization and how they were mitigated would add to the methodological rigour. How can the uncertainties from MODIS and Landsat-8 be quantified since Table 2 seems quite dependent on individual experiences?

Satellite imagery dates are reflected in Figure 3 and also included in all archived datasets. We have detailed our methods for analyzing accuracy of the manual digitization process

and reported those results in sections 3.4 and 3.5, so we are uncertain what additional information the reviewer is seeking. It is correct that individual experience influences digitization, so we had a single person complete all manual digitization as another means to reduce bias and minimize uncertainty. We feel this is the best possible method given the inherent constraints surrounding manual digitization. We also provide both written (Table 2) and visual (Figures 4-5) references to demonstrate ice classification categories. Early reports from external collaborators (personal communication) suggest good correlation for similar output from other research groups.

(7) Figure 10. What's the spatial resolution of frequencies in fast ice presence and glacial ice presence?

Based on our understanding of this request, we believe that the reviewer will find the information of interest in the stacked time series plots that are included in the primary manuscript and the appendix. Panel (b) in these figures shows the area and the percent fjord area coverage for fast ice and for the different glacier-derived ice types.

(8) I am slightly lost when you show solid ice discharge, landfast ice, and Glacier ice values. Can you explain their possible implications? Since, for me, they are just shown here without any inner interpretation or further analysis. The integration of diverse datasets is a strength of this study. However, discussing the challenges and uncertainties involved in combining different data types (e.g., remote sensing data with climate model outputs) and how these might affect the interpretation of results would be beneficial. Moreover, a more detailed analysis of the variability observed across the fjords and its potential biological implications would add depth to the study.

This study endeavors to provide characterization of the Southeast Greenland fjord ice environment but does not endeavor to provide in-depth analysis of the processes connecting these metrics (e.g., the transformation of solid ice discharge into glacier-derived ice with varying character and spatiotemporal presence). We hope that future research both inside and outside of our research team will continue to develop process, physical system, and biological insights that use these multiple data streams and move beyond the results presented and discussed in this paper.

(9) As I mentioned in the first point. The manuscript misses a deeper discussion on the specific biological applications of the physical environment characterization provided. For instance, detailing potential impacts on the habitat preferences, migration patterns, or population dynamics of key species would directly link the physical and biological aspects of the fjord systems.

We have intentionally limited the extent to which we speculate on biological implications of this physical system characterization since this study is a focus on the physical system observations and not the combining of physical and biological. We do biological context regarding the motivation for the study and also limited comment on biological implications within the Discussion and Conclusion. We prefer not to introduce additional speculation on biological implications because we are not providing new analysis on these connections.

---

## Author Comment (AC3)

Thank you to Dr. Nanna Karlsson for helpful comments that have improved the manuscript. The full content of comments and responses are below.

Reviewer Comments in **black** and responses in **red**

Comment from Nanna B. Karlsson

There are some overlaps between the dataset presented here and a dataset published last year (https://dataverse.geus.dk/dataset.xhtml?persistentId=doi:10.22008/FK2/BOVBVR), described in Karlsson et al., 2023 (https://geusbulletin.org/index.php/geusb/article/view/8338). The latter (which I will refer to as K2023) constrains the freshwater flux on a glacier-basin scale for all of Greenland but only considers the flux from the marine-terminating glaciers. As such, I consider this manuscript led by Moon as a natural and important step forward. I think it would be appropriate if the authors mention the improvement that their dataset offers compared to K2023 and why this step is necessary.

I am curious to know why the authors did not include the basally-derived melt in their dataset. This is included in the K2023 dataset and as such could easily be lifted from that dataset. The basal melt is a small but non-negligible freshwater term typically of the order of 5-10% of the total mass loss from the marine-terminating glaciers.

Best,

Nanna B. Karlsson
**Citation**: https://doi.org/10.5194/egusphere-2024-184-CC1

Thank you for bringing our attention to this complementary dataset. This prompted an expanded section 3.2 regarding the methods for freshwater flux, which now includes the definition of freshwater flux, discussion of the exclusion of in-fjord glacier-derived ice melt, and notes and discussion on the exclusion of basal melt. For the latter, we now note the complementary Karlsson et al. 2023 dataset and discuss how it or data from Karlsson et al. 2021 could be used to supplement our dataset and provide a rough estimate of the influence on results. Unfortunately, current funding for this research effort has ended and it was not possible for us to undertake sampling and full freshwater dataset revisions for this manuscript to add detailed basal melt to the final data products.